# REPA-FPO: A Fisher Policy Optimization for Efficient Flow Matching Training

**Tianyi Zheng\***  *zhengtianyi@vivo.com*
*BlueImage Lab, vivo Mobile Communication Co., Ltd*

**Fengxiang Yang\***  *yangfx@vivo.com*
*BlueImage Lab, vivo Mobile Communication Co., Ltd*

**Yijie Zhong**  *dun.haski@gmail.com*
*BlueImage Lab, vivo Mobile Communication Co., Ltd*

**Jiayang Gao**  *gjy0515@sjtu.edu.cn*
*Shanghai Jiao Tong University*

**Lv Tang**  *luckybird1994@gmail.com*
*Michael Smith Laboratories, University of British Columbia*

**Jinwei Chen**  *jinwei.chen@vivo.com*
*BlueImage Lab, vivo Mobile Communication Co., Ltd*

**Peng-Tao Jiang**  *pt.jiang@vivo.com*
*BlueImage Lab, vivo Mobile Communication Co., Ltd*

**Jia Wang**  *jiawang@sjtu.com*
*Shanghai Jiao Tong University*

**Bo Li[†]**  *libra@vivo.com*
*BlueImage Lab, vivo Mobile Communication Co., Ltd*

**Reviewed on OpenReview:** *https://openreview.net/forum?id=mRHipMopOC*

## Abstract

Flow Matching (FM) models are a leading class of generative models, widely used across diverse domains. However, FM models require large-scale training datasets, which makes training computationally expensive. Existing feature alignment (REPA) improves training efficiency but overlooks the role of the data itself, leaving further room for improvement. In this paper, we observe that different samples carry different amounts of Fisher information and thus contribute unequally to parameter learning in FM. This heterogeneity highlights the importance of accounting for sample-wise contributions during training. However, computing per-sample Fisher information accurately is prohibitively expensive in practice. To overcome this limitation, we provide a mathematical analysis showing that the loss magnitude can serve as an effective proxy for the trace of the Fisher Information Matrix (FIM), enabling efficient estimation. Building on this insight, we propose Fisher Policy Optimization (FPO), a strategy that dynamically reweights samples during training by shifting weight from low-FIM samples to high-FIM samples. Extensive experiments demonstrate that FPO improves both training efficiency and generation quality, while generalizing well across inference samplers, model architectures, and diffusion spaces.

---

[*]Equal Contribution.
[†]Corresponding Author

# 1 Introduction

Deep generative models have witnessed a paradigm shift with the development of Flow Matching models (FMs) Lipman et al. (2023), a training paradigm within the diffusion model (DM) family Ho et al. (2020); Song et al. (2021). FMs generate samples by progressively transporting a simple pre-defined distribution (e.g., noise) to the target data distribution via a probability flow (velocity) field, which is learned by training on stochastically interpolated noise–data pairs to predict the corresponding transport direction. As a result, FMs have established state-of-the-art performance across diverse modalities, with rapidly expanding applications in unconditional and conditional image synthesis Dhariwal & Nichol (2021); Rombach et al. (2022); Zheng et al. (2024a); Yang et al. (2026); Park et al. (2026); Lee et al. (2026) and video generation Chen et al. (2025). Consequently, improving the generative capability and training efficiency of FMs remains critical for real-world deployment.

To improve the generative capabilities of the model, extensive research has explored optimization from multiple perspectives. For example, LDM Rombach et al. (2022), DiT Peebles & Xie (2022), and SiT Ma et al. (2024), build a strong network architecture, while P2-Weight Choi et al. (2022), Min-SNR Hang et al. (2023), and ANT Go et al. (2024) re-weight loss function based on signal-to-noise (SNR). Moreover, BB-TDM Zheng et al. (2026) and SpeeD Wang et al. (2025a) refine timestep sampling, and recent works such as REPA Yu et al. (2025), HASTE Wang et al. (2025b) and REG Wu et al. (2025) achieve significant gains by aligning the features with pre-trained vision encoders (e.g., DINOv2 Oquab et al. (2023) and MAE He et al. (2021)). Despite these comprehensive advancements, a critical factor remains overlooked: the varying contribution of individual samples to the training process. Current design typically relies on randomly sampled batches and minimizes a **uniformly averaged loss**. However, our analysis reveals that the information density varies significantly across samples, rendering this naive averaging strategy sub-optimal as it dilutes the learning signal from the most informative data.

The training of FMs always relies on massive-scale datasets, such as the million-scale ImageNet Deng et al. (2009) and the billion-scale LAION Schuhmann et al. (2022). Within such vast samples, the information density of individual examples is intrinsically heterogeneous. Therefore, the naive uniform-averaging strategy over samples can waste substantial computation on low-information examples, thereby undermining training efficiency. While prior works, such as CEP Chen et al. (2024), attempt to enhance performance via conditional noise perturbations, DeltaFM Stoica et al. (2025) designs a sample-wise contrastive to prevent paths from intersecting in flow matching. They fall short of explicitly measuring the sample's importance. Recently, $D^2C$ Huang et al. (2025b) designs a framework to select the informative samples in the whole dataset. However, $D^2C$ relies on extra pre-trained models on the same data and introduces complex two-stage frameworks to select the informative samples, which limit its practical application. Given the tangible benefits already realized from such studies, quantifying the per-sample information content and dynamically adjusting the training process remain significant and unexplored challenges in FMs. Addressing this is pivotal for maximizing data efficiency and improving generative quality.

To address these challenges, we propose **Fisher Policy Optimization (FPO)**, a method that dynamically modulates sample contributions during FM training. To make sample informativeness comparable across various samples, we introduce a principled criterion grounded in classical statistics: the Fisher Information Matrix (FIM) Fisher (1925). In particular, we leverage the per-sample FIM as a metric to quantify how strongly each sample constrains the model parameters, thereby guiding our sample-wise reweighting policy. *Specifically, the FIM captures the local geometry of the parameter space, quantifying the local steepness of the loss manifold. This provides an effective measure of a sample's potential to induce substantial updates to the model parameters.* However, the explicit calculation of the FIM requires per-sample gradient evaluations, which incurs a prohibitive computational cost given the massive parameter space of modern FMs. To incorporate FIM into the training of the diffusion model, we establish the loss magnitude as a theoretically and computationally efficient proxy (in Proposition 3.1 and Proposition 3.2). Leveraging this proxy, FPO implements a gradient redistribution strategy within each training iteration, directing the model's focus toward high-information samples. We validate FPO by integrating it into different frameworks, including DiT Peebles & Xie (2022), SiT Ma et al. (2024), JiT Li & He (2025), REPA Yu et al. (2025), and REG Wu et al. (2025), with different prediction targets and generation tasks. Results demonstrate that FPO sig-

nificantly enhances generative quality and training efficiency. Furthermore, extensive experiments across diverse generative tasks and samplers confirm the robust generalizability of our method. We summarize our contributions

- To incorporate an efficient Fisher Information Matrix (FIM) estimate into the training of Flow Matching models, we derive a loss-based proxy that links the loss magnitude to the FIM. This proxy enables efficient, per-sample estimation of a relative Fisher information at each training iteration.

- Based on the aforementioned efficient FIM proxy, we propose Fisher Policy Optimization (FPO) to improve the training of FMs. FPO dynamically reweights and redistributes gradients across training samples according to their estimated relative Fisher information, encouraging the model to focus more on samples with higher Fisher Information. Compared with the previous average policy, this sample-adaptive design improves both training efficiency and generation quality.

- Our extensive experimental evaluation confirms that FPO is not only effective but also highly orthogonal to existing techniques. It delivers consistent improvements in generative quality and training speed across a wide range of frameworks and remains compatible with diverse advanced inference samplers.

## 2 Related Work

**Advancements in Diffusion Architectures and Training.** The development of diffusion models has been driven by significant architectural innovations. Foundational frameworks like LDM Rombach et al. (2022), DiT Peebles & Xie (2022), and SiT Ma et al. (2024) successfully scale diffusion processes to latent spaces and transformer backbones. Beyond architecture, substantial efforts focus on optimizing the training dynamics. Techniques such as P2 Choi et al. (2022), Min-SNR Hang et al. (2023), SpeeD Wang et al. (2025a), BB-TDM Zheng et al. (2026) and ANUT Kim et al. (2025b) introduce loss or timestep re-weight strategies to improve training efficiency. More recently, representation-alignment strategies (e.g., REPA Yu et al. (2025), REPA-E Leng et al. (2025), REG Wu et al. (2025), and iREPA Singh et al. (2025)) have emerged as a dominant paradigm. By aligning the internal features of diffusion backbones with pre-trained encoders (e.g., DINOv2 Oquab et al. (2023)), these methods accelerate training convergence and generation quality.

**Sample-Aware Optimization.** Despite these diverse improvements, a critical inefficiency remains prevalent across existing frameworks: the *uniform treatment* of training samples. Current methods optimize a simple average loss over randomly sampled batches, implicitly assuming equal contribution from every sample. This overlooks the heterogeneous information density inherent in large-scale datasets. While CEP Chen et al. (2024) and DeltaFM Stoica et al. (2025) partially mitigate this issue via conditional perturbations and a contrastive framework, they rely on heuristic randomness rather than a rigorous metric for quantifying sample information. $D^2C$ Huang et al. (2025b) proposes a two-stage framework for selecting informative samples, but relies on extra pre-trained models, limiting its practical applicability. Similarly, Focal Loss Lin et al. (2017) employs static mechanisms that cannot adaptively redistribute weights within each batch, and Importance Sampling Arouna (2004) incurs prohibitive cost due to per-sample gradient computation. A separate line of work deliberately reweights samples to *shift* the learned distribution. Energy-weighted Flow Matching (EFM) Zhang et al. (2025b) uses energy functions to bias generation in offline RL, while Reweighted Flow Matching via Unbalanced OT Song et al. (2025) derives weights from class frequencies for long-tailed generation. In both cases, the weights encode a pre-specified distributional bias and do not vanish as training converges. In contrast, FPO's weights are dynamic functions of the current per-sample loss; they serve to accelerate optimization rather than alter the convergence target, and self-anneal as training progresses. Moreover, d-TDA Mlodozeniec et al. (2025) proposes a post-hoc method for analyzing sample influence based on Hessian inverse and unrolled differentiation. In addition, adaptive policy learning for sample-wise importance has also been explored in multi-agent coordination Zhang et al. (2025a), where dynamic policies are learned to handle heterogeneous agent contributions. In contrast to the above, FPO provides an efficient, theoretically motivated method for dynamically measuring sample informativeness and can be seamlessly integrated into SOTA diffusion frameworks to enhance both training efficiency and generative quality.

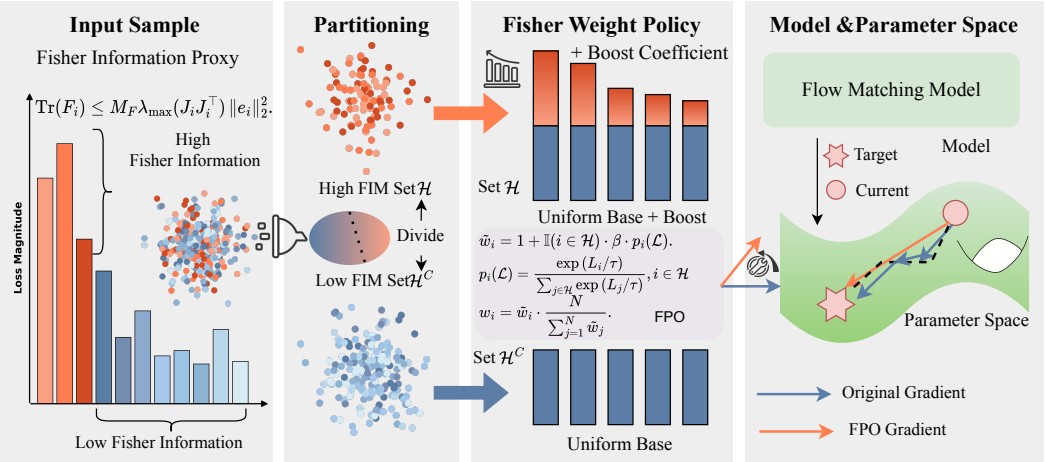

Figure 1: **The pipeline of FPO.** The core idea of FPO is to adaptively reweight sample gradients during training using the loss magnitude as a proxy for FIM, thereby amplifying the contribution of high Fisher information samples.

## 3 Method

### 3.1 Preliminaries

**Flow Matching.** Flow Matching Lipman et al. (2023); Liu et al. (2023) has become one of the most popular diffusion frameworks, often using Scalable Interpolant Transformers (SiT) Ma et al. (2024), which improve upon the DiT Peebles & Xie (2022). The Flow Matching constructs intermediate processes by linearly interpolating between the data and a Gaussian distribution. Specifically, it linearly combines the data $x_0$ with Gaussian noise $\epsilon$, as shown in equation 1:

$$\mathbf{x}_t = \alpha_t \mathbf{x}_0 + \sigma_t \epsilon \tag{1}$$

During the training stage, we train a neural network parameterized by $\theta$ to approximate the velocity field $\mathbf{v}(x_t, t) = \dot{x}_t$, defined as follows:

$$\begin{aligned} \mathbf{v}(\mathbf{x}, t) &= \mathbb{E}\left[\dot{\mathbf{x}}_t \mid \mathbf{x}_t = \mathbf{x}\right] \\ &= \dot{\alpha}_t \mathbb{E}\left[\mathbf{x}_0 \mid \mathbf{x}_t = \mathbf{x}\right] + \dot{\sigma}_t \mathbb{E}\left[\varepsilon \mid \mathbf{x}_t = \mathbf{x}\right]. \end{aligned} \tag{2}$$

Therefore, we minimize the following loss function in the training stage

$$\mathcal{L}_{\mathrm{v}}(\theta) = \int_0^T \mathbb{E}\left[\left\|\mathbf{v}_\theta\left(\mathbf{x}_t, t\right) - \dot{\alpha}_t \mathbf{x}_0 - \dot{\sigma}_t \varepsilon\right\|^2\right] \mathrm{d}t. \tag{3}$$

Upon completion of training, we generate samples by solving the inverse SDE Anderson (1982), which gradually transforms the prior Gaussian distribution into the data distribution. The specific formula is as follows:

$$\mathrm{d}\mathbf{X}_t = \mathbf{v}\left(\mathbf{X}_t, t\right) \mathrm{d}t - \frac{1}{2} w_t \mathbf{s}\left(\mathbf{X}_t, t\right) \mathrm{d}t + \sqrt{w_t}\, \mathrm{d}\overline{\mathbf{W}}_t \tag{4}$$

In equation 4, $\overline{\mathbf{W}}_t$ is a reverse-time Wiener process, $w_t$ is an arbitrary time-dependent diffusion coefficient, and $\mathbf{s}(\mathbf{x}, t)$ is the score can be expressed as $\mathbf{s}(\mathbf{x}, t) = -\sigma_t^{-1} \mathbb{E}\left[\varepsilon \mid \mathbf{x}_t = \mathbf{x}\right]$. We use $\mathbf{v}_\theta\left(\mathbf{x}_t, t\right)$ instead of $\mathbf{v}\left(\mathbf{x}_t, t\right)$ in the generation stage.

**Representation Alignment (REPA).** To enhance the feature representations of the Flow Matching, REPA Yu et al. (2025) proposes a feature alignment strategy. Specifically, REPA aims to accelerate training and improve generation quality by aligning the Flow Matching model's features with those of pre-trained

vision models (e.g., DINOv2 Oquab et al. (2023)) during training. REPA achieves this by introducing an MLP $h_\phi$ projection head that performs token-level alignment between features extracted by SiT and the pre-trained vision models, maximizing their similarity

$$\mathcal{L}_{\mathrm{REPA}}(\theta, \phi) := -\mathbb{E}_{\mathbf{x}_t, \epsilon, t}\left[\mathrm{sim}\left(F^n, h_\phi\left(H_t^{[n]}\right)\right)\right]. \tag{5}$$

In equation 5, $H_t^{[n]}$ is the output of the n-th SiT block, and $F^n$ is the n-th reference representation extracted by the pre-trained vision models. This alignment loss is jointly optimized with the Flow Matching loss during training to enhance both training efficiency and the generation quality of the Flow Matching model.

## 3.2   Analysis of Fisher Information in REPA

While the joint optimization of $\mathcal{L}_{\mathrm{v}}$ and $\mathcal{L}_{\mathrm{REPA}}$ defines the training objective, the standard training procedure relies on a fundamental, yet often overlooked, inefficiency: the naive uniform averaging of each sample. This design implicitly assumes that all samples in each training iteration contribute equally, which is generally sub-optimal. In practice, the contribution of informative samples can be diluted by a large number of low-information samples that the model already fits well. This observation motivates an adaptive training policy that amplifies the influence of informative samples. A key challenge is therefore to identify which samples are most informative. To this end, we turn to the principles of information geometry and the Fisher Information Matrix (FIM) Fisher (1925); Amari (2016). Intuitively, since the FIM is related to the expected outer product of per-sample gradients, it offers a way to quantify each sample's contribution.

**Empirical Fisher Information.** To quantify per-sample contributions from an optimization perspective, we adopt the *empirical Fisher Information Matrix* (empirical FIM) Kunstner et al. (2019), defined as the outer product of the per-sample negative log-likelihood gradient: $\mathbf{F}_i(\theta) = \nabla_\theta \ell_i(\theta) \nabla_\theta \ell_i(\theta)^\top$, where $\ell_i(\theta) = -\log p(y_i \mid x_i; \theta)$. The empirical FIM is a standard tool in optimization for measuring per-sample curvature and has been widely used as a stable positive semi-definite surrogate for the Hessian in high-dimensional models Martens (2020); Thomas et al. (2020). We note that the empirical FIM differs from the true generative Fisher Information Matrix $\mathbf{F}(\theta) = \mathbb{E}_{x \sim p_\theta}\left[\nabla_\theta \log p_\theta(x) \nabla_\theta \log p_\theta(x)^\top\right]$; as shown by Kunstner et al. (2019), the two coincide only at the global optimum. Our analysis operates entirely within the empirical Fisher framework, using it as a per-sample optimization diagnostic to identify informative samples, rather than claiming equivalence to the generative FIM. However, explicitly forming the per-sample empirical FIM is computationally prohibitive due to the $d \times d$ matrix size in parameter space. To obtain a tractable scalar summary, a common choice is the trace $\mathrm{Tr}(\mathbf{F}_i)$, which equals the sum of eigenvalues and captures the overall curvature induced by sample $x_i$ Martens (2020); Thomas et al. (2020); Huang et al. (2025a).

Next, we establish the relationship between $\mathcal{L}_{\mathrm{v}}$ and the FIM. The velocity prediction objective in Flow Matching minimizes a mean squared error (MSE) Lipman et al. (2023). For a single training sample $i$, we use

$$\mathcal{L}_{\mathrm{v}, i} = \|v_{\mathrm{true}, i} - v_{\theta, i}\|_2^2. \tag{6}$$

This objective admits an equivalent maximum-likelihood interpretation Shen et al. (2025) under a Gaussian observation model: assuming an isotropic Gaussian residual

$$v_{\mathrm{true}, i} = v_{\theta, i} + \varepsilon_i, \qquad \varepsilon_i \sim \mathcal{N}(0, \sigma^2 I), \tag{7}$$

we have $p(v_{\mathrm{true}, i} \mid v_{\theta, i}; \theta) = \mathcal{N}\left(v_{\mathrm{true}, i}; v_{\theta, i}, \sigma^2 I\right)$. The corresponding negative log-likelihood satisfies

$$-\log p(v_{\mathrm{true}, i} \mid v_{\theta, i}; \theta) = \frac{1}{2\sigma^2}\|v_{\mathrm{true}, i} - v_{\theta, i}\|_2^2 + \mathrm{C}, \tag{8}$$

Therefore, minimizing $\mathcal{L}_{\mathrm{v}, i}$ is equivalent to MLE up to an additive constant and a positive scaling factor. In the FIM derivation, we treat $\sigma^2$ as a fixed scalar; it controls the overall scale of the FIM but not its correlation structure.

We now establish the explicit relationship between the loss magnitude $e_i$ and a tractable per-sample Fisher information. Specifically, we adopt the empirical Fisher, defined as the outer product of the per-sample

negative log-likelihood gradient. Under the Gaussian observation model in equation 7, the empirical Fisher trace has the following form that couples the loss magnitude with the model Jacobian.

**Proposition 3.1** (Per-sample empirical FIM trace for FM). *Let $v_{\theta,i} \in \mathbb{R}^d$ be the model output for sample $i$ and $J_i = \frac{\partial v_{\theta,i}}{\partial \theta}$ is its Jacobian. Assume the Gaussian observation model in equation 7 with fixed variance, and define the error $e_i \triangleq v_{\theta,i} - v_{true,i}$. Let $\ell_i(\theta) \triangleq -\log p(v_{true,i} \mid \theta)$ and $g_i \triangleq \nabla_\theta \ell_i(\theta)$. Define the per-sample empirical Fisher term as $F_i \triangleq g_i g_i^\top$. Then the trace of the per-sample empirical Fisher satisfies*

$$\mathrm{Tr}(F_i) = \|g_i\|_2^2 = M_F \, e_i^\top \, (J_i J_i^\top) \, e_i, \tag{9}$$

*where $M_F > 0$ is a constant determined by the scaling of $\ell_i$ (i.e., the noise variance $\sigma^2$). Consequently, since $J_i J_i^\top \succeq 0$, we have $\mathrm{Tr}(F_i) \geq 0$, and*

$$0 \leq \mathrm{Tr}(F_i) \leq M_F \, \lambda_{\max}(J_i J_i^\top) \, \|e_i\|_2^2. \tag{10}$$

*Proof.* The proof is detailed in Appendix A.1. □

**Remark.** Proposition 3.1 links the per-sample empirical FIM trace to the loss magnitude via the $\mathrm{Tr}(F_i) \leq M_F \, \lambda_{\max} e_i^\top (J_i J_i^\top) e_i$. In particular, optimization practices (e.g., initialization, normalization, residual connections, and gradient clipping) tend to keep the local sensitivity numerically stable along training trajectories. This suggests that the Jacobian spectral norm $\|J_i(\theta)\|_2$ is often bounded in practice. Consequently, samples with vanishing loss ($e_i \to 0$) necessarily have vanishing empirical-Fisher contribution ($\mathrm{Tr}(F_i) \to 0$). *We emphasize that this does not assert the converse (large loss need not imply a large $\mathrm{Tr}(F_i)$); rather, near-zero loss is a sufficient indicator of negligible per-sample FIM contribution.* We empirically validate these findings in Figure 3, and the results in Figure 3 are consistent with our analysis.

Minimizing the REPA objective $\mathcal{L}_{\mathrm{REPA}}$ admits a directional maximum-likelihood interpretation under a von Mises-Fisher (vMF) model on the unit hypersphere Fisher (1953); Govindarajan et al. (2023); Taghia et al. (2014). For each sample $i$, let $f_i \in \mathbb{S}^{d-1}$ denote the normalized reference representation and let $v_{\theta,i} \in \mathbb{R}^d$ be the predicted representation with normalized direction $\mu_{\theta,i} = v_{\theta,i}/\|v_{\theta,i}\|_2 \in \mathbb{S}^{d-1}$. The vMF likelihood is

$$p(f_i \mid x_i; \theta) = C_d(\kappa) \exp\big(\kappa \, f_i^\top \mu_{\theta,i}\big), \tag{11}$$

where $\kappa > 0$ is the concentration parameter and $C_d(\kappa)$ is the normalization constant. The log-likelihood is

$$\log p(f_i \mid x_i; \theta) = \kappa \, f_i^\top \mu_{\theta,i} + \log C_d(\kappa). \tag{12}$$

With the per-sample REPA loss defined as $\mathcal{L}_{\mathrm{REPA},i} = -f_i^\top \mu_{\theta,i}$, we obtain $\log p(f_i \mid x_i; \theta) = -\kappa \mathcal{L}_{\mathrm{REPA},i} + \log C_d(\kappa)$, since $\log C_d(\kappa)$ is independent of $\theta$, the score satisfies $\nabla_\theta \log p(f_i \mid x_i; \theta) = -\kappa \nabla_\theta \mathcal{L}_{\mathrm{REPA},i}$. Consequently, the per-sample empirical Fisher term induced by the REPA can be written as $F_i = \kappa^2 \nabla_\theta \mathcal{L}_{\mathrm{REPA},i} \nabla_\theta \mathcal{L}_{\mathrm{REPA},i}^\top$, which provides a direct connection between the REPA gradient and the Fisher information.

**Proposition 3.2** (Per-sample REPA FIM Trace). *For sample $i$, let $v_{\theta,i}^n \in \mathbb{R}^d$ be the block output and $J_i^n = \frac{\partial v_{\theta,i}^n}{\partial \theta} \in \mathbb{R}^{d \times |\theta|}$ its Jacobian. The unit direction is*

$$\mu_{\theta,i}^n = \frac{v_{\theta,i}^n}{\|v_{\theta,i}^n\|_2} \in \mathbb{S}^{d-1}. \tag{13}$$

*Let $f_i^n \in \mathbb{S}^{d-1}$ be a normalized reference representation of the block $n$, and assume positive alignment $(\mu_{\theta,i}^n)^\top f_i^n > 0$.*

$$\Pi_\mu \triangleq I - \mu\mu^\top, \quad \tilde{e}_i^n \triangleq \frac{1}{\|v_{\theta,i}^n\|_2} \, \Pi_{\mu_{\theta,i}^n} f_i^n. \tag{14}$$

*Consider the vMF directional model with fixed concentration $\kappa > 0$,*

$$p(f_i^n \mid x_i; \theta) \propto \exp\big(\kappa (f_i^n)^\top \mu_{\theta,i}^n\big). \tag{15}$$

*Then the per-sample empirical FIM trace satisfies*

$$\mathrm{Tr}(F_i^n) = M_F \, (\tilde{e}_i^n)^\top \big(J_i^n (J_i^n)^\top\big) \tilde{e}_i^n, \qquad M_F = \kappa^2. \tag{16}$$

*Consequently, since $J_i^n (J_i^n)^\top \succeq 0$,*

$$0 \le \mathrm{Tr}(F_i^n) \le M_F \, \lambda_{\max}\big(J_i^n (J_i^n)^\top\big) \, \|\tilde{e}_i^n\|_2^2. \tag{17}$$

*Proof.* The proof is detailed in Appendix A.2. $\qquad\square$

**Remark.** Proposition 3.2 also bounds the per-sample FIM trace by equation 17. The error term $\|\tilde{e}_i^n\|_2$ represents the magnitude of the target's projection onto the subspace *orthogonal* to the prediction $\mu_{\theta,i}^n$. Under the positive alignment condition (i.e., $(\mu_{\theta,i}^n)^\top f_i^n > 0$), this projection error is strictly monotonic with the REPA loss: as the loss approaches its minimum of $-1$ (perfect alignment), the orthogonal error $\|\tilde{e}_i^n\|_2$ vanishes to 0.

**Unified Conclusion.** Based on the insights from Proposition 3.1 and 3.2, we obtain a consistent conclusion: samples with low loss magnitude contribute negligibly to the per-sample empirical FIM trace. We also empirically validated this conclusion in section 4.3. This motivates prioritizing samples with larger loss magnitude, using the loss as a lightweight proxy to identify samples that are more likely to drive meaningful parameter updates.

---

**Algorithm 1** Training of FPO.

---

**Require:** Dataset $q(x_0)$, Model parameters $\theta$, MLP $h_\phi$.
1: **while** Not Converged **do**
2:      Batch data $\mathcal{B} = \{x_i\}_{i=1}^N \sim q(x_0)$.
3:      $t \sim \mathcal{U}[0,1], \boldsymbol{\epsilon} \sim \mathcal{N}(\mathbf{0}, \boldsymbol{I})$, Encodered $F^n$.
4:      $\mathbf{x}_t = (1-t)\mathbf{x} + t\epsilon$, $z_t = h_\phi(x_t)$
5:      Compute $\mathcal{L}_v$ (3) and $\mathcal{L}_{\mathrm{REPA}}$ (5) of each sample.
6:      Fisher Policy Optimization (FPO) with Algorithm 2
7: **end while**
8: Trained Model parameters $\theta$.

---

**Algorithm 2** Fisher Policy Optimization (FPO)

---

1: **Input:** per-sample losses $\mathcal{L}_v = \{L_i(\theta)\}_{i=1}^N$ and $\mathcal{L}_{\mathrm{REPA}} = \{L_i(\phi)\}_{i=1}^N$
2: **Hyperparameters:** Retention ratio $r$, Temperature $\tau$, Augmentation factor $\beta$, Time interval $\mathcal{T}$.
3: Sort $\mathcal{L}_v, \mathcal{L}_{\mathrm{REPA}} \in \mathcal{T}$ separately in each set: $L_{h(1)} \ge L_{h(2)} \ge \cdots \ge L_{h(N)}$ .
4: Identify high FIM subset: $\mathcal{H} \leftarrow \{h(1), \ldots, h(\lfloor r \cdot N \rfloor)\}$
5: Initialize raw weights: $\tilde{w} \leftarrow \mathbf{1} \in \mathbb{R}^N$
6: **for** each sample $i \in \mathcal{H}$ **do**
7:      Compute relative importance: $p_i \leftarrow \dfrac{\exp(L_i/\tau)}{\sum_{j \in \mathcal{H}} \exp(L_j/\tau)}$
8:      Apply intensity boost: $\tilde{w}_i \leftarrow 1 + \beta \cdot p_i$
9: **end for**
10: Normalize weights: $w_i \leftarrow \tilde{w}_i \cdot \dfrac{N}{\sum_{j=1}^N \tilde{w}_j}$
11: Compute weighted objective: $\mathcal{J}_{\mathrm{FPO}} \leftarrow \frac{1}{N} \sum_{i=1}^N w_i L_i$
12: Update: $\theta \leftarrow \theta - \eta \nabla_\theta \mathcal{J}_{\mathrm{FPO}}$, $h_\phi \leftarrow h_\phi - \eta \nabla_\phi \mathcal{J}_{\mathrm{FPO}}$

---

### 3.3 Fisher Policy Optimization

Building on the analysis above, we find that the per-sample loss magnitude correlates with sample informativeness (e.g., the FIM trace), making it a simple and computationally lightweight proxy that can be

used in practical REPA Yu et al. (2025); Wu et al. (2025) training. Therefore, instead of naive averaging, which assigns equal weight to all samples, we perform per-iteration adaptive resource allocation, dynamically down-weighting low-information samples and up-weighting high-FIM ones, which reduces ineffective updates and improves overall training efficiency.

Accordingly, we design **Fisher Policy Optimization (FPO)**, a plug-in policy that instantiates the above per-iteration adaptive allocation for REPA via gradient re-weighting. Here, policy refers to a sample-level weight allocation strategy that dynamically distributes gradient budgets within each batch, rather than a sequential decision policy in the reinforcement learning sense Yang et al. (2026); Zhang et al. (2025a). As illustrated in Figure 1, FPO computes normalized sample weights from the informativeness estimates and uses them to modulate each step's gradients.

Let $\mathcal{B} = \{x_i\}_{i=1}^N$ be a minibatch with per-sample losses $\mathcal{L} = (L_1, \ldots, L_N)^\top \in \mathbb{R}^N$. Using loss magnitude as a proxy for sample FIM, we define the high-information set $\mathcal{H} \subseteq \{1, \ldots, N\}$ as the indices of the top-$M$ losses, where $M = \lfloor rN \rfloor$. Within $\mathcal{H}$, we assign a Softmax policy

$$p_i(\mathcal{L}) \triangleq \begin{cases} \dfrac{\exp(L_i/\tau)}{\sum_{j \in \mathcal{H}} \exp(L_j/\tau)}, & i \in \mathcal{H}, \\ 0, & i \notin \mathcal{H}, \end{cases} \tag{18}$$

where $\tau > 0$ controls the concentration of the allocation. We then allocate the boost coefficient $\beta \geq 0$ over $\mathcal{H}$ according to $p_i$, yielding the unnormalized weights

$$\tilde{w}_i \triangleq 1 + \mathbb{I}(i \in \mathcal{H})\,\beta\,p_i(\mathcal{L}). \tag{19}$$

Finally, to decouple FPO from the learning-rate adjustment, we normalize the weights to keep the *total* weight fixed.

$$w_i \triangleq \tilde{w}_i \cdot \frac{N}{\sum_{j=1}^N \tilde{w}_j}, \qquad \text{s.t.,} \sum_{i=1}^N w_i = N. \tag{20}$$

This normalization makes FPO a pure *redistribution* mechanism: it changes how gradient contributions are allocated across samples without introducing an implicit learning-rate adjustment. In particular, the extra weight budget is concentrated on $\mathcal{H}$ and distributed according to $p_i(\mathcal{L})$, where $\tau$ controls the concentration within $\mathcal{H}$ and $\beta$ controls the overall strength of the boosting.

**Practical Consideration.** Previous research Wang et al. (2025b) finds that the REPA loss function primarily provides overall feature alignment during the late diffusion stage ($t \to 1$). Therefore, we apply FPO to the REPA loss during this stage to amplify the gain of the REPA regularization (i.e., $t \in \mathcal{T}[0.9, 1]$) and align with the positive alignment condition. *It is worth noting that the use of FPO for the two items is separate.* Other than that, FPO is applied identically to both $\mathcal{L}_v$ and $\mathcal{L}_{\text{REPA}}$. Therefore, we uniformly use $\mathcal{J}_{FPO}$ for subsequent analysis. The final training procedure of FPO is summarized in Algorithm 1 and 2.

### 3.4 Discussion and Theoretical Analysis

While FPO reallocates the gradient budget via sample re-weighting, its optimization behavior goes beyond previous re-weight design. For example, P2-weight Choi et al. (2022) treat the weights as *SNR-Based constant* in the training stage. In contrast, FPO keeps the policy $w(\mathcal{L}(\theta))$ differentiable with respect to $\theta$, making the weights dynamic functions of the current per-sample losses and introducing an additional gradient pathway through $\nabla_\theta w$. This extra cost is minimal: FPO introduces only $M = |\mathcal{H}| = \lfloor rN \rfloor$ additional learnable parameters and one Top-$M$ operation over the $N$ per-sample losses per iteration. In terms of complexity, the extra overhead is $\mathcal{O}(N \log M)$ for Top-$M$ selection, which is independent of the model parameter size. In practice, their cost is negligible compared to the backward passes required to compute $\{\nabla_\theta L_i\}_{i=1}^N$ for the REPA model with multiple parameters.

To further elucidate the mechanism behind FPO, we analyze the gradient dynamics of the FPO objective.

**Theorem 3.3** (Gradient decomposition of FPO). *Consider each iteration of FPO contains $N$ samples with per-sample losses $L_i(\theta)$, and the FPO objective is $\mathcal{J}_{\text{FPO}}(\theta) = \frac{1}{N}\sum_{i=1}^{N} w_i(\theta)\, L_i(\theta)$. Conditioned on the FPO, $L_i(\theta)$ and $w_i(\theta)$ are differentiable with respect to $\theta$. Then the gradient of $\mathcal{J}_{\text{FPO}}$ with respect to model parameters $\theta$ is $\nabla_\theta \mathcal{J}_{\text{FPO}} = \frac{1}{N}\sum_{i=1}^{N} w_i \nabla_\theta L_i + \frac{1}{N}\sum_{i=1}^{N} L_i \nabla_\theta w_i$. For a ratio $r \in (0,1]$ and let the high-information set contain exactly $M = \lfloor rN \rfloor$ samples. Re-index the high-information set elements as $\mathcal{H} = \{h_1, \ldots, h_M\}$, Let $p_i$ denote the Softmax policy over $\mathcal{H}$ and let $w_i$ denote the corresponding normalized weights, both defined in Sec. 3.3. Therefore, for any sample $i$, the gradient of the weight $w_i(\theta)$ with respect to $\theta$ can be expressed as:*

$$\nabla_\theta w_i = \frac{\beta}{\frac{1}{N}\sum_{k=1}^{N}(1+\beta p_k)} \nabla_\theta p_i$$
$$- \frac{1+\beta p_i}{\left(\frac{1}{N}\sum_{k=1}^{N}(1+\beta p_k)\right)^2} \frac{1}{N}\sum_{k=1}^{N}\beta \nabla_\theta p_k. \tag{21}$$

*where*

$$\nabla_\theta p_i = \begin{cases} \frac{1}{\tau} p_i \left(\nabla_\theta L_i - \sum_{m=1}^{M} p_{h_m}\nabla_\theta L_{h_m}\right), & i \in \mathcal{H}, \\ 0, & i \in \mathcal{H}^C. \end{cases} \tag{22}$$

*Proof.* The proof is detailed in Appendix A.3. $\qquad\square$

Based on Theorem 3.3, we derive the following implications for analyzing FPO.

**(i) Gradient Components.** The FPO update decomposes into a *loss-gradient* term $\frac{1}{N}\sum_{i=1}^{N} w_i \nabla_\theta L_i$ and an additional *weight-gradient* term $\frac{1}{N}\sum_{i=1}^{N} L_i \nabla_\theta w_i$. The former rescales the per-sample training signal, while the latter introduces an additional regular term.

**(ii) The weight-gradient term is driven by the high-information set.** Since the policy $p_i$ is defined over the high-information set $\mathcal{H}$, we have $\nabla_\theta p_i = 0$ for $i \in \mathcal{H}^C$. Consequently, the gradients of $w_i(\theta)$ are induced primarily by the high-information samples through $\nabla_\theta p_i$.

**(iii) Relative emphasis within $\mathcal{H}$.** For $i \in \mathcal{H}$, $\nabla_\theta p_i$ depends on $\nabla_\theta L_i - \sum_{m=1}^{M} p_{h_m}\nabla_\theta L_{h_m}$, i.e., the deviation of a sample's loss gradient from the policy-weighted average over $\mathcal{H}$. This indicates that the weight-gradient term reallocates optimization emphasis among high-information samples.

Beyond the gradient decomposition above, an important question is whether FPO's reweighting introduces persistent bias compared to standard uniform training. The following proposition shows that FPO preserves the global optimum, and its gradient bias vanishes near convergence.

**Proposition 3.4** (Convergence Consistency). *Let $\mathcal{J}_{\text{FPO}}(\theta) = \frac{1}{N}\sum_{i=1}^{N} w_i(\theta)L_i(\theta)$ and $\mathcal{J}_{\text{uni}}(\theta) = \frac{1}{N}\sum_{i=1}^{N} L_i(\theta)$, where $L_i \geq 0$ and $w_i > 0$ with $\sum_{i=1}^{N} w_i = N$. Let $G = \max_i \|\nabla_\theta L_i\|$ denote the maximum per-sample loss gradient norm, and let $W = \frac{1}{N}\sum_i \|\nabla_\theta w_i\|$ denote the average weight gradient norm. Define the gradient bias $B(\theta) = \nabla_\theta \mathcal{J}_{\text{FPO}} - \nabla_\theta \mathcal{J}_{\text{uni}}$. Then:*

**(i)** *The global minima coincide: $\mathcal{J}_{\text{FPO}}(\theta) = 0 \Leftrightarrow \mathcal{J}_{\text{uni}}(\theta) = 0$.*

**(ii)** *If $L_i(\theta) \to 0$ for all $i$, then $\|B(\theta)\| \to 0$.*

*Proof.* The proof is detailed in Appendix A.4. $\qquad\square$

**Remark.** Proposition 3.4 establishes two key properties. (i) guarantees that FPO and uniform training share the same global optimum. (ii) shows that as per-sample losses vanish, FPO's gradient bias also vanishes and the optimization dynamics recover those of standard uniform training. This formally shows FPO's self-annealing behavior: FPO modifies the optimization trajectory but not the destination.

# 4 Experiments

## 4.1 Experimental Setup

**Implementation details.** We adopt the REPA Yu et al. (2025) and the stronger REG Wu et al. (2025) as our baselines. Experiments are performed on ImageNet-1K (256 and 512) Deng et al. (2009). We adopt FID Heusel et al. (2017), sFID Nash et al. (2021), IS, Precision, and Recall Kynkäänniemi et al. (2019) as our evaluation metrics, generating 50K samples for their computation. During inference, unless otherwise specified, we default to using the SDE-250 sampler. More implementation details are in the Appendix B.

## 4.2 Comparative Evaluation

**Quantitative results.** In Table 1, we compare the performance of FPO across different architectures. FPO comprehensively enhances the generative capabilities of different REPA architectures, demonstrating significant improvements in both IS and FID, while also improving or maintaining Precision and Recall. Furthermore, FPO can be integrated with CFG to enhance the performance of conditional generation further.

**Training Speed.** As analyzed above, FPO effectively enhances both convergence speed and generation quality, as shown in Figures 2 and 6, where the results are evaluated across multiple metrics and model scales. Specifically, Figure 2 shows that FPO yields significant and consistent improvements across various architectures in terms of training speed and generation quality, demonstrating that the proposed method generalizes well beyond a single model configuration.

Table 1: Comparison of different REPA models on ImageNet-256 with (*) and without CFG.

| Method | IS↑ | FID↓ | Prec.↑ | Rec.↑ |
|---|---|---|---|---|
| REPA-B | 59.90 | 24.40 | 0.59 | 0.65 |
| **REPA-B + FPO** | **65.96** | **22.50** | **0.59** | **0.65** |
| REPA-B* | 173.61 | 5.59 | 0.78 | 0.53 |
| **REPA-B* + FPO** | **175.55** | **5.38** | **0.79** | **0.53** |
| REPA-L | 109.20 | 10.00 | 0.69 | 0.65 |
| **REPA-L + FPO** | **112.30** | **9.76** | **0.68** | **0.66** |
| REPA-L* | 264.34 | 2.81 | 0.85 | 0.55 |
| **REPA-L* + FPO** | **269.16** | **2.73** | **0.85** | **0.55** |
| REPA-XL | 122.60 | 7.90 | 0.70 | 0.65 |
| **REPA-XL + FPO** | **128.33** | **7.58** | **0.70** | **0.65** |
| REPA-XL* | 281.89 | 1.93 | 0.82 | 0.59 |
| **REPA-XL* + FPO** | **283.03** | **1.90** | **0.82** | **0.60** |

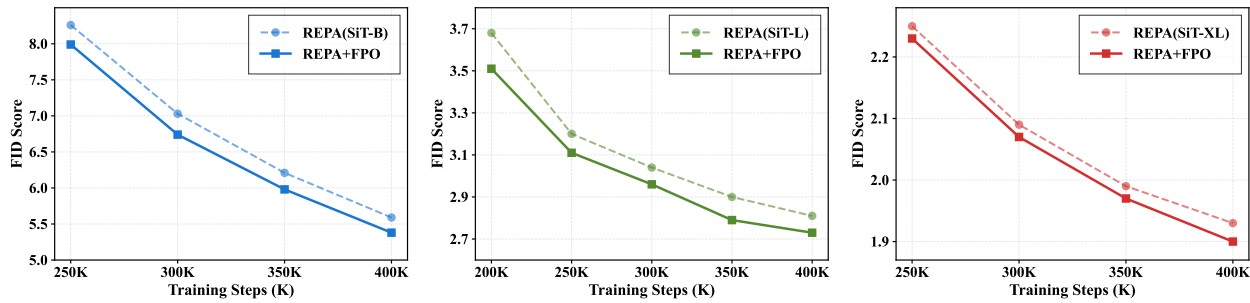

Figure 2: **Quantitative Comparison.** Convergence comparison on ImageNet between REPA and REPA-FPO. REPA-FPO achieves the fastest convergence and better FID across different model sizes (B, L and XL).

**Orthogonality of FPO.** Since our FPO performs intra-batch relative adjustments based on sample-level FIM proxies, it can be seamlessly integrated with other designs. Recently, REG Wu et al. (2025) introduce a CLS token to provide stronger semantic guidance, making it the current SOTA. In Table 2, we integrate FPO with REG to further investigate the orthogonality of FPO. The results demonstrate that FPO further improves the generation quality of REG, achieving an FID below *2.0* with lightweight SiT-L (400K), surpassing larger models such as DiT-XL, SiT-XL and MaskDiT Zheng et al. (2024b) trained with 7M. Moreover, SiT-XL (REG) + FPO achieves superior generation quality, outperforming other methods across most metrics.

Table 2: **Quantitative Comparison.** Comparison of various state-of-the-art models on the ImageNet-256 benchmark with (w) and without (w/o) CFG. By integrating the most advanced REG method, FPO achieves comprehensive improvements in most metrics.

| Method | Iter. | Generation w/o guidance | | | | | Generation w guidance | | | | |
|---|---|---|---|---|---|---|---|---|---|---|---|
| | | FID↓ | sFID↓ | IS↑ | Prec.↑ | Rec.↑ | FID↓ | sFID↓ | IS↑ | Prec.↑ | Rec.↑ |
| *Latent Diffusion Transformer without REPA* | | | | | | | | | | | |
| DiT-XL | 7M | 9.62 | 6.85 | 121.50 | 0.67 | 0.67 | 2.27 | 4.60 | 278.20 | 0.83 | 0.57 |
| SiT-XL | 7M | 8.26 | 6.32 | 131.65 | 0.68 | 0.67 | 2.06 | 4.50 | 270.30 | 0.82 | 0.59 |
| MaskDiT | 7M | 5.69 | 10.34 | 177.99 | 0.74 | 0.60 | 2.28 | 5.67 | 276.56 | 0.80 | 0.61 |
| *Latent Diffusion Transformer with REPA* | | | | | | | | | | | |
| SiT-L (REPA) | 400K | 10.00 | 5.20 | 109.20 | 0.69 | 0.65 | 2.81 | 4.88 | 264.34 | **0.85** | 0.55 |
| SiT-XL (REPA) | 400K | 7.90 | 5.06 | 112.60 | 0.70 | 0.65 | 1.93 | 4.59 | 281.89 | 0.82 | 0.59 |
| SiT-XL (REPA-E) | 400K | 3.46 | **4.17** | 159.80 | **0.77** | 0.63 | 1.67 | **4.12** | 266.30 | 0.80 | 0.63 |
| SiT-L (REG) | 400K | 4.60 | 5.21 | 167.60 | 0.75 | 0.63 | 2.05 | 4.74 | 261.65 | 0.77 | 0.63 |
| SiT-XL (REG) | 400K | 3.40 | 4.87 | 184.10 | 0.76 | 0.64 | 1.71 | 4.65 | 280.90 | 0.78 | 0.63 |
| SiT-L (REG)+FPO | 400K | 4.31 | 5.14 | 170.98 | 0.75 | 0.64 | 1.99 | 4.71 | 266.74 | 0.78 | 0.63 |
| **SiT-XL (REG)+FPO** | 400K | **2.70** | 4.90 | **201.80** | 0.76 | **0.64** | 1.67 | 4.62 | **293.39** | 0.78 | **0.64** |

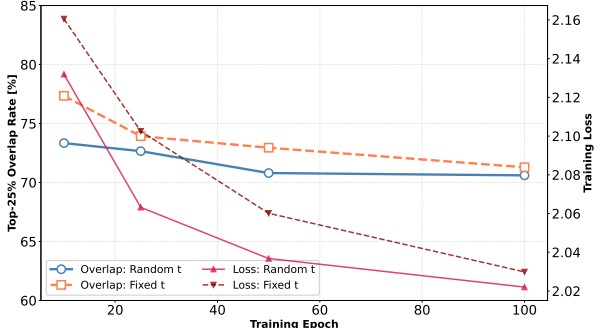

Figure 3: **Overlap Rate Analysis.** Evaluation of the alignment between high-FIM and high-loss samples throughout the training.

Figure 4: Effect of FPO on training loss compared to REPA. The green area indicates that FPO improves the prediction results.

**Further Results of FPO on More Training Iterations.** To further evaluate FPO, we train the state-of-the-art REG framework with FPO for more iterations. The results are summarized in Table 3. FPO consistently improves performance in both unconditional and conditional generation. Notably, under conditional generation (w/ CFG), FPO achieves an FID of 1.46 with only 800K training iterations, outperforming REG trained for 1.5M iterations.

**Different Diffusion Space.** To further validate FPO's generalizability, we evaluate it in pixel space based on JiT Li & He (2025). Experiments are conducted on JiT-B-16 and JiT-B-32, trained for 200 epochs. For inference, we use the 50-step Heun sampler Heun et al. (1900); Li & He (2025) with CFG. As shown in Table 4, FPO consistently improves performance, demonstrating strong versatility.

**Comparison with Other Reweighting Strategies.** To further validate the design of FPO, we compare it against several representative reweighting strategies on SiT-B (REPA) trained for 400K steps on ImageNet-256. Specifically, we consider: (1) P2-Weight Choi et al. (2022), a static reweighting method based on (SNR), applied to both the denoising and REPA losses jointly (Denoise+REPA) and to the denoising loss only (Denoise-only); (2) Hard Sample, which applies a uniform boost to the top-$M$ highest-loss samples without softmax differentiation; and (3) Adaptive Non-Uniform Timestep Sampling Kim et al. (2025a;b), which reweights training based on timestep difficulty. The results are summarized in Table 5. FPO outperforms all compared methods. P2-Weight is a static design with predefined weights based on SNR, which cannot adaptively balance the REPA alignment loss and the denoising loss—applying it to both losses jointly leads to significant degradation (FID 30.09). Hard Sample applies uniform boost to top-$M$ samples without softmax differentiation; its inferiority to FPO confirms that the softmax-based policy within $\mathcal{H}$ is a meaningful design

Table 3: Training efficiency and generation quality comparison under unconditional (w/o CFG) and conditional (w/ CFG) settings.

| Method | Uncond. (w/o CFG) | | Cond. (w/ CFG) | |
|---|---|---|---|---|
| | Iters | FID ↓ | Iters | FID ↓ |
| Vanilla | 1M | 2.70 | 1.5M | 1.48 |
| + FPO | 800K$_{(1.25\times)}$ | **2.66** | 800K$_{(1.88\times)}$ | **1.46** |

Table 4: **Quantitative Comparison.** Comparison of FID score on JiT model in the pixel space across different resolution ImageNet (256×256, 512×512) datasets.

| Method | JiT-B-16 (256) | JiT-B-32 (512) |
|---|---|---|
| Vanilla | 4.63 | 5.55 |
| + FPO | **4.59** | **5.48** |

choice. The timestep-based Adaptive Non-Uniform strategy also improves over the baseline, but remains below FPO, demonstrating the effectiveness of per-sample reweighting.

Table 5: Comparison with different reweighting strategies on SiT-B. FPO outperforms all compared methods in both IS and FID.

| Method | IS↑ | FID↓ |
|---|---|---|
| REPA (Baseline) | 59.90 | 24.40 |
| P2-Weight (Denoise+REPA) | 53.10 | 30.09 |
| P2-Weight (Denoise-only) | 62.50 | 25.37 |
| Hard Sample (Uniform boost) | 64.56 | 22.99 |
| Adaptive Non-Uniform (Timestep) | 64.34 | 22.74 |
| **FPO (Ours)** | **65.96** | **22.50** |

Table 6: Ablation studies of FPO on ImageNet. (a) Effect of high-FIM subset ratio $r\%$; (b) Effect of augmentation $\beta$.

| (a) High-Information Ratio | | | (b) Augmentation $\beta$ | | |
|---|---|---|---|---|---|
| Ratio $r\%$ | IS↑ | FID↓ | Aug $\beta$ | IS↑ | FID↓ |
| Baseline | 59.90 | 24.40 | Baseline | 59.90 | 24.40 |
| 60% | 65.82 | 22.70 | $\beta=1$ | **66.11** | 22.77 |
| 80% | **65.96** | **22.50** | $\beta=2$ | 65.96 | **22.50** |
| 100% | 64.24 | 23.41 | $\beta=5$ | 66.00 | 22.61 |

### 4.3 Analysis and Discussion

**Empirical Validation.** To empirically validate the consistency between FIM and loss magnitude, we conduct a toy experiment with a simple MLP trained on a 2D dataset via FM, where the per-sample FIM trace can be computed exactly. As shown in Figure. 3, while the loss decreases monotonically during training, the overlap between the top-loss samples and the top-FIM samples remains stable more than 70%. To eliminate potential confounding from the timestep $t$, we consider two settings: (i) sampling an independent random $t$ for each sample, and (ii) using a fixed shared $t$ for all samples. The overlap is consistent in both settings, supporting our assumption that high-loss samples tend to be high-FIM in Proposition 3.1 and 3.2.

**Effectiveness of FPO.** In Figure 4, we compare models trained with REPA and with FPO, evaluating their prediction errors on the same data. The results show that FPO performs adaptive sample reweighting guided by FIM proxy, thereby increasing the influence of informative samples while reducing that of samples with low information. This leads to consistently lower average loss and reduced prediction error. To further quantify this effect, we conduct a detailed per-sample loss comparison on 1,024 fixed evaluation samples (SiT-XL), as shown in Table 7. FPO achieves lower mean denoising loss at all checkpoints. The improvements on hard samples are larger in magnitude than the marginal degradation on easy samples, consistent with FPO's design. For the REPA projection loss, FPO shows advantage in earlier training and gradually converges with the baseline—consistent with HASTE Wang et al. (2025b) showing that the REPA loss produces gradient conflicts late in training. In Figure 5, we also compare the generation results. In the low-FIM set, FPO does not suffer from degraded generation quality due to gradient redistribution. In contrast, in the more complex high-FIM set, FPO produces substantially higher-quality samples.

**Effects of $r$ and $\beta$.** We evaluate the robustness of FPO with respect to its hyperparameters. As shown in Table 6, FPO consistently improves over the vanilla model under a wide range of settings. As $r$ and $\beta$ increase, performance initially improves but then degrades, suggesting that overly large values may destabilize training and lead to suboptimal results. Overall, FPO outperforms vanilla training across diverse hyperparameter choices, demonstrating its robustness.

## 5 Conclusion

In this paper, we propose FPO, which dynamically allocates training gradients by estimating the Fisher information of different samples, enabling more efficient training of FMs. We first analyze the rationale and

Table 7: Per-sample loss comparison between REPA (Baseline) and FPO on SiT-XL. $\Delta = \text{Baseline} - \text{FPO}$; positive values indicate FPO achieves lower loss.

| | Denoising Loss | | Proj Loss | |
|---|---|---|---|---|
| Step | $\Delta$ (B−F) | FPO Better% | $\Delta$ (B−F) | FPO Better% |
| 250K | +0.047% | 53.4% | +0.030% | 51.7% |
| 300K | +0.044% | 54.4% | +0.003% | 51.6% |
| 350K | +0.049% | 51.7% | −0.002% | 49.9% |
| 400K | +0.050% | 53.7% | −0.027% | 49.5% |

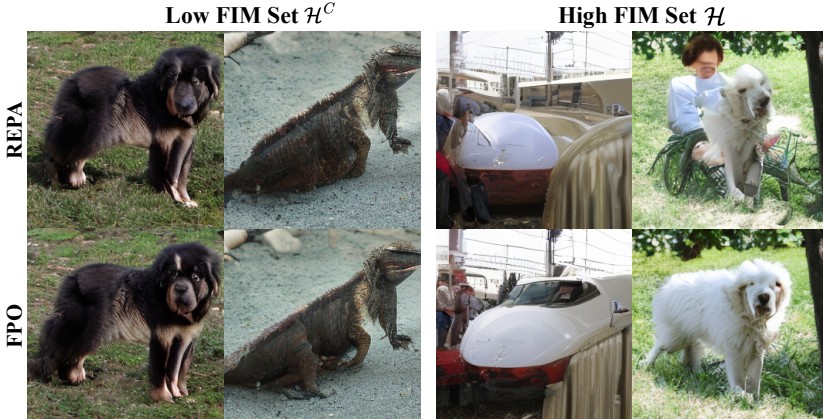

**Low FIM Set $\mathcal{H}^C$**    **High FIM Set $\mathcal{H}$**

Figure 5: **Qualitative Comparison.** The generation results from REPA and FPO on the ImageNet.

efficiency of using the loss magnitude as a proxy for the FIM and provide empirical evidence to support this design. We further validate FPO across several SOTA frameworks, including REPA and REG. Extensive experiments show that FPO substantially improves training efficiency and generative quality for FMs, while remaining robust across different inference samplers, model architectures, and diffusion spaces.

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

## A   Appendix

## Appendix Overview

In the appendix, we first provide detailed proofs for the theoretical results in Section A. We then give detailed experimental settings in Section B. Section C presents additional analysis and discussion of our method, while Section D includes further visual results.

## A   Detailed Proof

### A.1   Proof of Proposition 3.1

We first prove the following lemma:

**Lemma (A.1).** *Let $M \in \mathbb{R}^{d \times d}$ be a real symmetric matrix. Then for any nonzero vector $x \in \mathbb{R}^d$, the Rayleigh quotient*

$$R_M(x) \triangleq \frac{x^\top M x}{x^\top x} \tag{23}$$

*satisfies*

$$\lambda_{\min}(M) \ \leq \ R_M(x) \ \leq \ \lambda_{\max}(M), \tag{24}$$

*where $\lambda_{\min}(M)$ and $\lambda_{\max}(M)$ denote the smallest and largest eigenvalues of $M$. Equivalently,*

$$\lambda_{\min}(M) \|x\|_2^2 \ \leq \ x^\top M x \ \leq \ \lambda_{\max}(M) \|x\|_2^2. \tag{25}$$

*Proof.* Since $M$ is real symmetric, it admits an orthogonal eigendecomposition $M = Q\Lambda Q^\top$ where $Q^\top Q = I$ and $\Lambda = \mathrm{diag}(\lambda_1, \ldots, \lambda_d)$ with $\lambda_k \in \mathbb{R}$. For any nonzero $x$, let $z = Q^\top x$. Then $\|z\|_2 = \|x\|_2$ and

$$x^\top M x = x^\top Q \Lambda Q^\top x = z^\top \Lambda z = \sum_{k=1}^d \lambda_k z_k^2, \qquad x^\top x = z^\top z = \sum_{k=1}^d z_k^2. \tag{26}$$

Hence

$$R_M(x) = \frac{x^\top M x}{x^\top x} = \frac{\sum_{k=1}^d \lambda_k z_k^2}{\sum_{k=1}^d z_k^2} = \sum_{k=1}^d \lambda_k \underbrace{\frac{z_k^2}{\sum_{j=1}^d z_j^2}}_{\triangleq w_k}. \tag{27}$$

The weights $w_k$ satisfy $w_k \geq 0$ and $\sum_{k=1}^d w_k = 1$, so $R_M(x)$ is a convex combination of $\{\lambda_k\}_{k=1}^d$. Therefore,

$$\min_k \lambda_k \leq R_M(x) \leq \max_k \lambda_k, \tag{28}$$

i.e., $\lambda_{\min}(M) \ \leq \ R_M(x) \ \leq \ \lambda_{\max}(M)$. Multiplying by $x^\top x = \|x\|_2^2$ yields the equivalent quadratic-form bounds. $\square$

**Proposition (3.1** Per-sample empirical FIM trace)**.** *Let $v_{\theta,i} \in \mathbb{R}^d$ be the model output for sample $i$ and $J_i = \frac{\partial v_{\theta,i}}{\partial \theta}$ its Jacobian. Assume the Gaussian observation model in equation 7 with fixed variance, and define the error $e_i \triangleq v_{\theta,i} - v_{true,i}$. Let $\ell_i(\theta) \triangleq -\log p(v_{true,i} \mid \theta)$ and $g_i \triangleq \nabla_\theta \ell_i(\theta)$. Define the per-sample empirical Fisher term as $F_i \triangleq g_i g_i^\top$. Then*

$$\mathrm{Tr}(F_i) = M_F \, e_i^\top (J_i J_i^\top) e_i, \tag{29}$$

*where $M_F > 0$ is a scaling constant determined by the scaling of $\ell_i$ (i.e., by $\sigma^2$ under equation 7). Consequently, since $J_i J_i^\top \succeq 0$, we have $\mathrm{Tr}(F_i) \geq 0$, and*

$$0 \leq \mathrm{Tr}(F_i) \leq M_F \, \lambda_{\max}(J_i J_i^\top) \|e_i\|_2^2. \tag{30}$$

*Proof.* Under the Gaussian observation model in equation 7,

$$p(v_{\text{true},i} \mid \theta) = \mathcal{N}(v_{\text{true},i}; v_{\theta,i}, \sigma^2 I). \tag{31}$$

The negative log-likelihood is

$$\ell_i(\theta) \triangleq -\log p(v_{\text{true},i} \mid \theta) = \frac{1}{2\sigma^2}\|v_{\text{true},i} - v_{\theta,i}\|_2^2 + C = \frac{1}{2\sigma^2}\|e_i\|_2^2 + C, \tag{32}$$

where $C$ is independent of $\theta$ and $e_i \triangleq v_{\theta,i} - v_{\text{true},i}$. Since $\frac{\partial e_i}{\partial \theta} = \frac{\partial v_{\theta,i}}{\partial \theta} = J_i$, The per-sample gradient is

$$g_i \triangleq \nabla_\theta \ell_i(\theta) = \frac{1}{2\sigma^2}\nabla_\theta(e_i^\top e_i) = \frac{1}{\sigma^2} J_i^\top e_i. \tag{33}$$

By definition, the per-sample empirical Fisher term is $F_i \triangleq g_i g_i^\top$, hence

$$F_i = \frac{1}{\sigma^4} J_i^\top e_i e_i^\top J_i. \tag{34}$$

Taking the trace and using $\text{Tr}(uu^\top) = \|u\|_2^2$ with $u = J_i^\top e_i$ yields

$$\text{Tr}(F_i) = \|g_i\|_2^2 = \frac{1}{\sigma^4}\|J_i^\top e_i\|_2^2 = \frac{1}{\sigma^4}e_i^\top(J_i J_i^\top)e_i. \tag{35}$$

Let $M_F \triangleq \sigma^{-4} > 0$, so equation 35 gives $\text{Tr}(F_i) = M_F e_i^\top(J_i J_i^\top)e_i$. Moreover, $J_i J_i^\top$ is symmetric positive semidefinite (i.e., $J_i J_i^\top \succeq 0$), so $\text{Tr}(F_i) \geq 0$. Applying Lemma A.1 with $M = J_i J_i^\top$ and $x = e_i$ gives

$$e_i^\top(J_i J_i^\top)e_i \leq \lambda_{\max}(J_i J_i^\top)\|e_i\|_2^2, \tag{36}$$

and multiplying by $M_F$ yields the upper bound. $\qquad\square$

## A.2 Proof of Proposition 3.2

**Proposition** (**3.2** Per-sample REPA FIM Trace). *For sample $i$, let $v_{\theta,i}^n \in \mathbb{R}^d$ be the block output and $J_i^n = \frac{\partial v_{\theta,i}^n}{\partial \theta} \in \mathbb{R}^{d \times |\theta|}$ its Jacobian. The unit direction is*

$$\mu_{\theta,i}^n = \frac{v_{\theta,i}^n}{\|v_{\theta,i}^n\|_2} \in \mathbb{S}^{d-1}. \tag{37}$$

*Let $f_i^n \in \mathbb{S}^{d-1}$ be a normalized reference representation of the block $n$, and assume positive alignment $(\mu_{\theta,i}^n)^\top f_i^n > 0$.*

$$\Pi_\mu \triangleq I - \mu\mu^\top, \qquad \tilde{e}_i^n \triangleq \frac{1}{\|v_{\theta,i}^n\|_2}\Pi_{\mu_{\theta,i}^n} f_i^n. \tag{38}$$

*Consider the vMF directional model with fixed concentration $\kappa > 0$,*

$$p(f_i^n \mid x_i; \theta) \propto \exp\big(\kappa(f_i^n)^\top \mu_{\theta,i}^n\big). \tag{39}$$

*Then the per-sample empirical FIM trace satisfies*

$$\text{Tr}(F_i^n) = M_F (\tilde{e}_i^n)^\top \big(J_i^n (J_i^n)^\top\big)\tilde{e}_i^n, \qquad M_F = \kappa^2. \tag{40}$$

*Consequently, since $J_i^n (J_i^n)^\top \succeq 0$,*

$$0 \leq \text{Tr}(F_i^n) \leq M_F \lambda_{\max}\big(J_i^n (J_i^n)^\top\big)\|\tilde{e}_i^n\|_2^2. \tag{41}$$

*Proof.* Recall the vMF log-likelihood

$$\log p(f_i^n \mid x_i; \theta) = \kappa (f_i^n)^\top \mu_{\theta,i}^n + C, \tag{42}$$

where $C$ is independent of $\theta$ since $\kappa$ is fixed.

Let $v = v_{\theta,i}^n$ and $\mu = \mu_{\theta,i}^n = v/\|v\|_2$ (with $\|v\|_2 > 0$). The Jacobian of normalization is

$$\frac{\partial \mu}{\partial v} = \frac{1}{\|v\|_2} \left(I - \mu \mu^\top\right) = \frac{1}{\|v\|_2} \Pi_\mu. \tag{43}$$

Thus, with $J_i^n = \frac{\partial v_{\theta,i}^n}{\partial \theta}$,

$$\nabla_\theta \log p(f_i^n \mid x_i; \theta) = \kappa (J_i^n)^\top \nabla_v \left((f_i^n)^\top \mu\right) = \kappa (J_i^n)^\top \left(\frac{1}{\|v\|_2} \Pi_\mu f_i^n\right) = \kappa (J_i^n)^\top \tilde{e}_i^n. \tag{44}$$

Define $F_i^n = ss^\top$ with $s = \nabla_\theta \log p(f_i^n \mid x_i; \theta)$. Then

$$F_i^n = \kappa^2 (J_i^n)^\top \tilde{e}_i^n (\tilde{e}_i^n)^\top J_i^n. \tag{45}$$

Taking the trace and using $\mathrm{Tr}(J^\top aa^\top J) = a^\top (JJ^\top)a$ yields

$$\mathrm{Tr}(F_i^n) = \kappa^2 (\tilde{e}_i^n)^\top \left(J_i^n (J_i^n)^\top\right) \tilde{e}_i^n. \tag{46}$$

Finally, Applying Lemma A.1 gives

$$0 \leq \mathrm{Tr}(F_i^n) \leq \kappa^2 \, \lambda_{\max}\!\left(J_i^n (J_i^n)^\top\right) \|\tilde{e}_i^n\|_2^2. \tag{47}$$

$\square$

## A.3   Proof of Theorem 3.3

**Theorem** (**3.3** Gradient decomposition of FPO). *Consider each iteration of FPO contains $N$ samples with per-sample losses $L_i(\theta)$, and the FPO objective is $\mathcal{J}_{\mathrm{FPO}}(\theta) = \frac{1}{N} \sum_{i=1}^N w_i(\theta) L_i(\theta)$. Conditioned on the FPO, $L_i(\theta)$ and $w_i(\theta)$ are differentiable with respect to $\theta$. Then the gradient of $\mathcal{J}_{\mathrm{FPO}}$ with respect to model parameters $\theta$ is $\nabla_\theta \mathcal{J}_{\mathrm{FPO}} = \frac{1}{N} \sum_{i=1}^N w_i \nabla_\theta L_i + \frac{1}{N} \sum_{i=1}^N L_i \nabla_\theta w_i$. For a ratio $r \in (0,1]$ and let the high-information set contain exactly $M = \lfloor rN \rfloor$ samples. Re-index the high-information set elements as $\mathcal{H} = \{h_1, \ldots, h_M\}$, Let $p_i$ denote the Softmax policy over $\mathcal{H}$ and let $w_i$ denote the corresponding normalized weights, both defined in Sec. 3.3. Therefore, for any sample $i$, the gradient of the weight $w_i(\theta)$ with respect to $\theta$ can be expressed as:*

$$\nabla_\theta w_i = \frac{\beta}{\frac{1}{N} \sum_{k=1}^N (1 + \beta p_k)} \nabla_\theta p_i - \frac{1 + \beta p_i}{\left(\frac{1}{N} \sum_{k=1}^N (1 + \beta p_k)\right)^2} \frac{1}{N} \sum_{k=1}^N \beta \nabla_\theta p_k, \tag{48}$$

*where*

$$\nabla_\theta p_i = \begin{cases} \dfrac{1}{\tau} p_i \left(\nabla_\theta L_i - \displaystyle\sum_{m=1}^M p_{h_m} \nabla_\theta L_{h_m}\right), & i \in \mathcal{H}, \\[4mm] 0, & i \in \mathcal{H}^C. \end{cases} \tag{49}$$

*Proof.* By the product rule, we have the following decomposition

$$\nabla_\theta \mathcal{J}_{\mathrm{FPO}} = \nabla_\theta \left(\frac{1}{N} \sum_{i=1}^N w_i L_i\right) = \frac{1}{N} \sum_{i=1}^N \left(w_i \nabla_\theta L_i + L_i \nabla_\theta w_i\right), \tag{50}$$

Next we derive $\nabla_\theta p_i$ and $\nabla_\theta w_i$. Since the $\mathcal{H} = \{h_1, \ldots, h_M\}$ and the Softmax policy over $\mathcal{H}$ is given by

$$p_i = \begin{cases} \dfrac{\exp(L_i/\tau)}{\sum_{m=1}^{M} \exp(L_{h_m}/\tau)}, & i \in \mathcal{H}, \\ 0, & i \in \mathcal{H}^C, \end{cases}$$

For $i \in \mathcal{H}$, the Softmax Jacobian restricted to $\mathcal{H}$ is

$$\frac{\partial p_i}{\partial L_j} = \frac{1}{\tau} p_i (\delta_{ij} - p_j), \qquad j \in \mathcal{H}. \tag{51}$$

Applying the chain rule,

$$\nabla_\theta p_i = \sum_{j \in \mathcal{H}} \frac{\partial p_i}{\partial L_j} \nabla_\theta L_j = \frac{1}{\tau} p_i \left( \nabla_\theta L_i - \sum_{j \in \mathcal{H}} p_j \nabla_\theta L_j \right) = \frac{1}{\tau} p_i \left( \nabla_\theta L_i - \sum_{m=1}^{M} p_{h_m} \nabla_\theta L_{h_m} \right), \tag{52}$$

which is the first case of equation 49. For the second case $i \in \mathcal{H}^C$, $p_i \equiv 0$ by definition, hence $\nabla_\theta p_i = 0$, which prove equation 49. Then define the normalized weights by

$$w_i \triangleq \frac{1 + \beta p_i}{Z}, \qquad Z \triangleq \frac{1}{N} \sum_{k=1}^{N} (1 + \beta p_k) \qquad (\beta \geq 0).$$

By the quotient rule,

$$\nabla_\theta w_i = \frac{\beta \nabla_\theta p_i}{Z} - \frac{1 + \beta p_i}{Z^2} \nabla_\theta Z. \tag{53}$$

Moreover,

$$\nabla_\theta Z = \frac{1}{N} \sum_{k=1}^{N} \beta \nabla_\theta p_k. \tag{54}$$

Substituting the expression of $\nabla_\theta Z$ yields equation 48. Moreover, we note that $\sum_{k=1}^{N} p_k = \sum_{i \in \mathcal{H}} p_i = 1$ implies $\sum_{k=1}^{N} \nabla_\theta p_k = 0$, so $\nabla_\theta Z = 0$ and hence $\nabla_\theta w_i = (\beta/Z) \nabla_\theta p_i$ in FPO. $\qquad \square$

### A.4 Proof of Proposition 3.4

**Proposition** (3.4 Convergence Consistency). *Let $\mathcal{J}_{\text{FPO}}(\theta) = \frac{1}{N} \sum_{i=1}^{N} w_i(\theta) L_i(\theta)$ and $\mathcal{J}_{\text{uni}}(\theta) = \frac{1}{N} \sum_{i=1}^{N} L_i(\theta)$, where $L_i \geq 0$ and $w_i > 0$ with $\sum_{i=1}^{N} w_i = N$. Let $G = \max_i \|\nabla_\theta L_i\|$ denote the maximum per-sample loss gradient norm, and let $W = \frac{1}{N} \sum_i \|\nabla_\theta w_i\|$ denote the average weight gradient norm. Define the gradient bias $B(\theta) = \nabla_\theta \mathcal{J}_{\text{FPO}} - \nabla_\theta \mathcal{J}_{\text{uni}}$. Then:*

*(i) The global minima coincide: $\mathcal{J}_{\text{FPO}}(\theta) = 0 \Leftrightarrow \mathcal{J}_{\text{uni}}(\theta) = 0$.*

*(ii) If $L_i(\theta) \to 0$ for all $i$, then $\|B(\theta)\| \to 0$.*

*Proof.* **(i)** Since $w_i > 0$ and $L_i \geq 0$, we have $\sum_i w_i L_i = 0$ if and only if $L_i = 0$ for all $i$, which is identical to the condition $\sum_i L_i = 0$.

**(ii)** By Theorem 3.3, the gradient bias decomposes as:

$$B(\theta) = \underbrace{\frac{1}{N} \sum_{i=1}^{N} (w_i - 1) \nabla_\theta L_i}_{B_1} + \underbrace{\frac{1}{N} \sum_{i=1}^{N} L_i \nabla_\theta w_i}_{B_2}. \tag{55}$$

Let $\epsilon = \max_i L_i$ and $\delta = \max_{i,j} |L_i - L_j|$. For $B_2$, since $L_i \leq \epsilon$, we have $\|B_2\| \leq \epsilon \cdot W = O(\epsilon)$. For $B_1$, when $|L_i - L_j| \leq \delta$, the softmax policy satisfies $|p_i - 1/M| = O(\delta/\tau)$, which implies $|w_i - 1| = O(\delta/\tau)$, hence $\|B_1\| \leq G \cdot O(\delta/\tau)$. When $L_i \to 0$ for all $i$, both $\epsilon \to 0$ and $\delta \to 0$, therefore $\|B(\theta)\| \to 0$. $\qquad \square$

# B    Detailed Experimental Setup

**Hyperparameter setup.**  In Table 8, we present the hyperparameter configurations of different model scales. To ensure a fair comparison, we follow the hyperparameters used in previous work Yu et al. (2025). During inference, unless otherwise specified, we default to using the SDE sampler for 250 inference steps. For DiT Zheng et al. (2024a), SiT Ma et al. (2024), REG  Wu et al. (2025), and JiT Li & He (2025), we also follow the default settings without making special modifications.

Table 8: Hyperparameter settings of FPO across different model scales.

| Backbone | SiT-B | SiT-L | SiT-XL |
|---|---|---|---|
| **Architecture** | | | |
| #Params | 132M | 460M | 677M |
| Input | $32 \times 32 \times 4$ | $32 \times 32 \times 4$ | $32 \times 32 \times 4$ |
| Layers | 12 | 24 | 28 |
| Hidden dim. | 768 | 1,024 | 1,152 |
| Num. heads | 12 | 16 | 16 |
| **REPA settings** | | | |
| $\lambda$ | 0.5 | 0.5 | 0.5 |
| Alignment depth | 4 | 8 | 8 |
| $\mathrm{sim}(\cdot, \cdot)$ | cos. sim. | cos. sim. | cos. sim. |
| Encoder $\mathcal{E}_{VF}(I)$ | DINOv2-B | DINOv2-B | DINOv2-B |
| **Optimization** | | | |
| Batch size | 256 | 256 | 256 |
| Optimizer | AdamW | AdamW | AdamW |
| lr | 0.0001 | 0.0001 | 0.0001 |
| $(\beta_1, \beta_2)$ | (0.9, 0.999) | (0.9, 0.999) | (0.9, 0.999) |
| **Interpolants** | | | |
| $\alpha_t$ | $1 - t$ | $1 - t$ | $1 - t$ |
| $\sigma_t$ | $t$ | $t$ | $t$ |
| $w_t$ | $\sigma_t$ | $\sigma_t$ | $\sigma_t$ |
| Training objective | v-prediction | v-prediction | v-prediction |
| Sampler | Euler-Maruyama | Euler-Maruyama | Euler-Maruyama |
| Sampling steps | 250 | 250 | 250 |

# C    Further Experimental Results

**Further Analysis and Comparison.**  Previous research CEP Chen et al. (2024) improves conditional generation by randomly perturbing per-sample condition embedding, which can be viewed as a stochastic way of injecting redistribution into training. However, CEP performs unsatisfactorily in unconditional generation tasks. This result also suggests an important takeaway: different samples contribute unequal amounts of useful information during optimization, and treating them uniformly (or randomly redistributing) can be suboptimal. We compare CEP and FPO on FFHQ Karras et al. (2021) in Table 9. All experiments use a DiT-B Peebles & Xie (2022) with noise prediction (i.e., $\epsilon$-prediction). In inference, we employ a 100-step SDE sampler and generate 50K samples for evaluation. The results show that FPO consistently improves the baseline, and FPO can also be combined with CEP to further enhance performance. These findings support that FPO is effective across different prediction targets and exhibits strong plug-and-play compatibility with existing designs.

**Vanilla Flow Matching Model.**  Vanilla Flow Matching models Lipman et al. (2023) do not rely on feature alignment and are trained only with the flow matching loss. To validate the FPO without REPA, we

Table 9: **Quantitative Comparison.** FPO is applicable to noise prediction and is compatible with other re-weight methods.

| Unconditional FFHQ (256×256). | | | | | | | | |
|---|---|---|---|---|---|---|---|---|
| | DiT-B w/o FPO | | | | DiT-B w/ FPO | | | |
| Method | FID↓ | sFID↓ | Prec.↑ | Rec.↑ | FID↓ | sFID↓ | Prec.↑ | Rec.↑ |
| Vanilla | 10.18 | 10.03 | 0.67 | 0.46 | 10.07 | 9.97 | 0.67 | 0.46 |
| + CEP | 11.68 | 11.18 | 0.65 | 0.45 | **9.85** | **9.91** | **0.68** | **0.46** |

further evaluate it on CelebA-HQ using a SiT-B backbone trained without REPA. For inference, we use a 100-step SDE sampler and generate 10K samples for evaluation. As shown in Table 10, FPO improves both generation quality and training efficiency, demonstrating its effectiveness beyond REPA.

Table 10: **Quantitative Comparison.** FPO can also be directly applied in the vanilla flow matching models.

| Iterations | 50K | 80K | 90K | 100K |
|---|---|---|---|---|
| SiT-B | 10.86 | 7.05 | 6.73 | 6.48 |
| + FPO | **10.63** | **6.69** | **6.49** | **6.33** |

**Robustness for Different Fast Samplers.** In practical applications, in addition to commonly used SDE samplers, ODE samplers are also widely adopted. To further validate the generalizability of FPO, we evaluate it under different samplers and varying numbers of function evaluations (NFEs). The results are reported in Table 11. FPO consistently yields significant performance gains across all sampler configurations, demonstrating its robustness.

Table 11: **Quantitative Comparison.** FPO improves generation quality consistently across ODE and SDE samplers with different NFE.

| ODE&SDE NFE | IS↑ | FID↓ | Prec.↑ | Rec.↑ |
|---|---|---|---|---|
| ODE-100 (Vanilla) | 61.77 | 25.11 | 0.58 | 0.65 |
| **ODE-100 (FPO)** | **62.61** | **24.49** | **0.58** | **0.65** |
| ODE-250 (Vanilla) | 61.63 | 24.75 | 0.58 | 0.65 |
| **ODE-250 (FPO)** | **62.31** | **24.12** | **0.58** | **0.65** |
| SDE-100 (Vanilla) | 63.98 | 24.29 | 0.59 | 0.65 |
| **SDE-100 (FPO)** | **65.40** | **23.65** | **0.59** | **0.65** |
| SDE-250 (Vanilla) | 59.90 | 24.40 | 0.59 | 0.65 |
| **SDE-250 (FPO)** | **65.96** | **22.50** | **0.59** | **0.65** |

**Text-to-Image Generation.** To further validate the generalizability of FPO, we evaluate it on COCO Lin et al. (2014) with the U-ViT Bao et al. (2023) architecture for text-to-image generation. As shown in Table 12, FPO achieves a lower FID at fewer iterations, demonstrating its effectiveness in other domains.

**Stability Evaluation.** To further assess the effectiveness of FPO, we evaluate REPA-XL with classifier-free guidance (CFG) under multiple random seeds. Table 13 reports the FID results. Across all different seeds, FPO achieves consistently lower FID than the baseline. Moreover, FPO has a smaller variance and a tighter max-min range of FID, which suggests that FPO exhibits better stability.

Table 12: **Quantitative Comparison.** Text-to-image generation on COCO with U-ViT. FPO achieves lower FID with fewer training iterations.

| Method | Iterations | FID↓ |
|---|---|---|
| U-ViT (Baseline) | 1000K | 5.95 |
| U-ViT + FPO | 800K | **5.82** |

Table 13: **Quantitative Comparison.** Stability of FID comparison across different random seeds based on REPA-XL with CFG.

| Method | Seed-1 | Seed-2 | Seed-3 | Average |
|---|---|---|---|---|
| Vanilla | 1.93 | 1.97 | 1.95 | $1.95 \pm 0.02$ |
| + FPO | **1.90** | **1.92** | **1.92** | **$1.91 \pm 0.01$** |

**More Metric Speed Evaluation.** Furthermore, we also evaluate other metrics of FPO in Figure 6. The result in Figure 6 demonstrates that FPO effectively accelerates convergence and improves the final performance across various generation metrics, highlighting the generalizability of FPO.

**More Qualitative Comparison.** By comparing the images generated by REG-XL and FPO in Figure 7, we observe that FPO improves generation quality by avoiding many distortions and reducing artifacts such as warping. Moreover, FPO produces more realistic textures and finer details, resulting in images with noticeably better naturalness and visual fidelity.

## D    More Visualization Results

In this section, we provide additional unconditional generation results in Figure 8 and conditional generation results in Figure 9. We employ the 250-step SDE sampler in the inference stage.

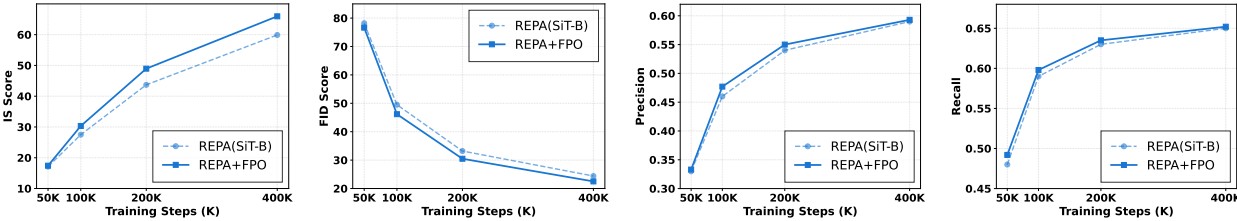

Figure 6: Comparison of convergence speed across different metrics. FPO consistently accelerates convergence for all evaluated metrics, demonstrating its effectiveness in improving training efficiency.

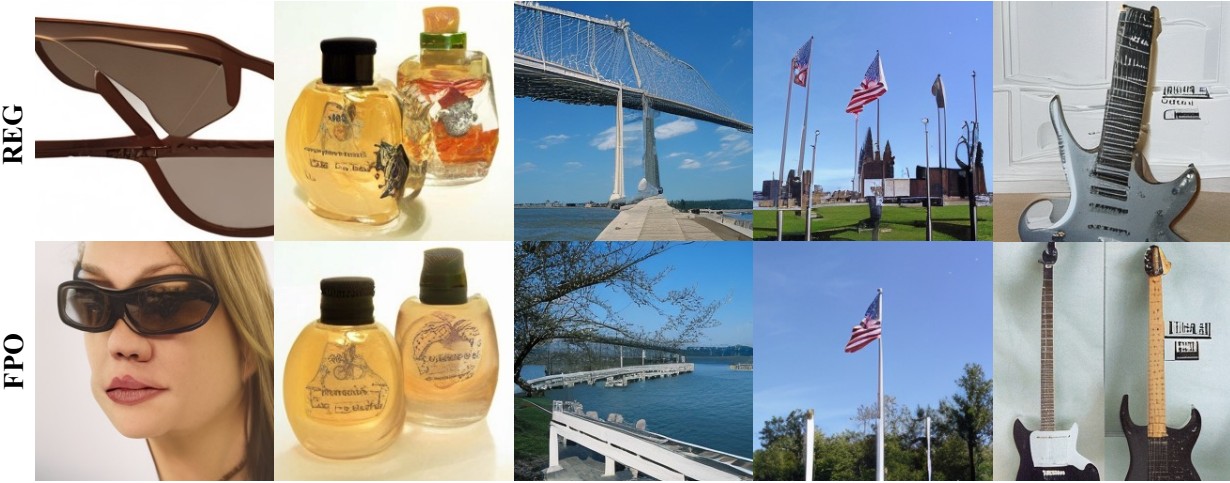

Figure 7: **Qualitative Comparison.** The generation results from REG (SiT-XL) and FPO on the ImageNet dataset with CFG.

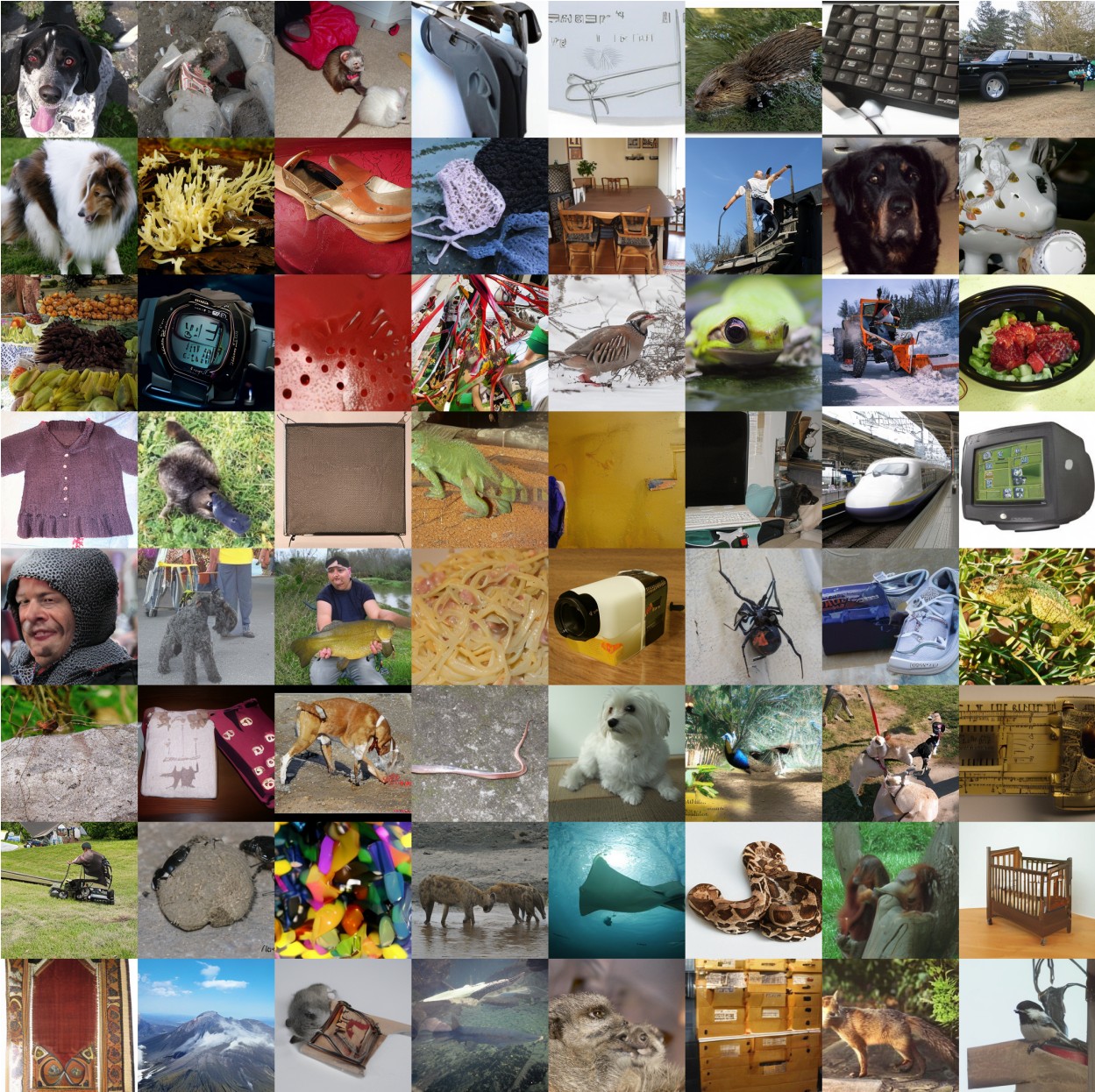

Figure 8: **Qualitative Comparison.** The generation results from REG (SiT-XL) with FPO on the ImageNet dataset without CFG.

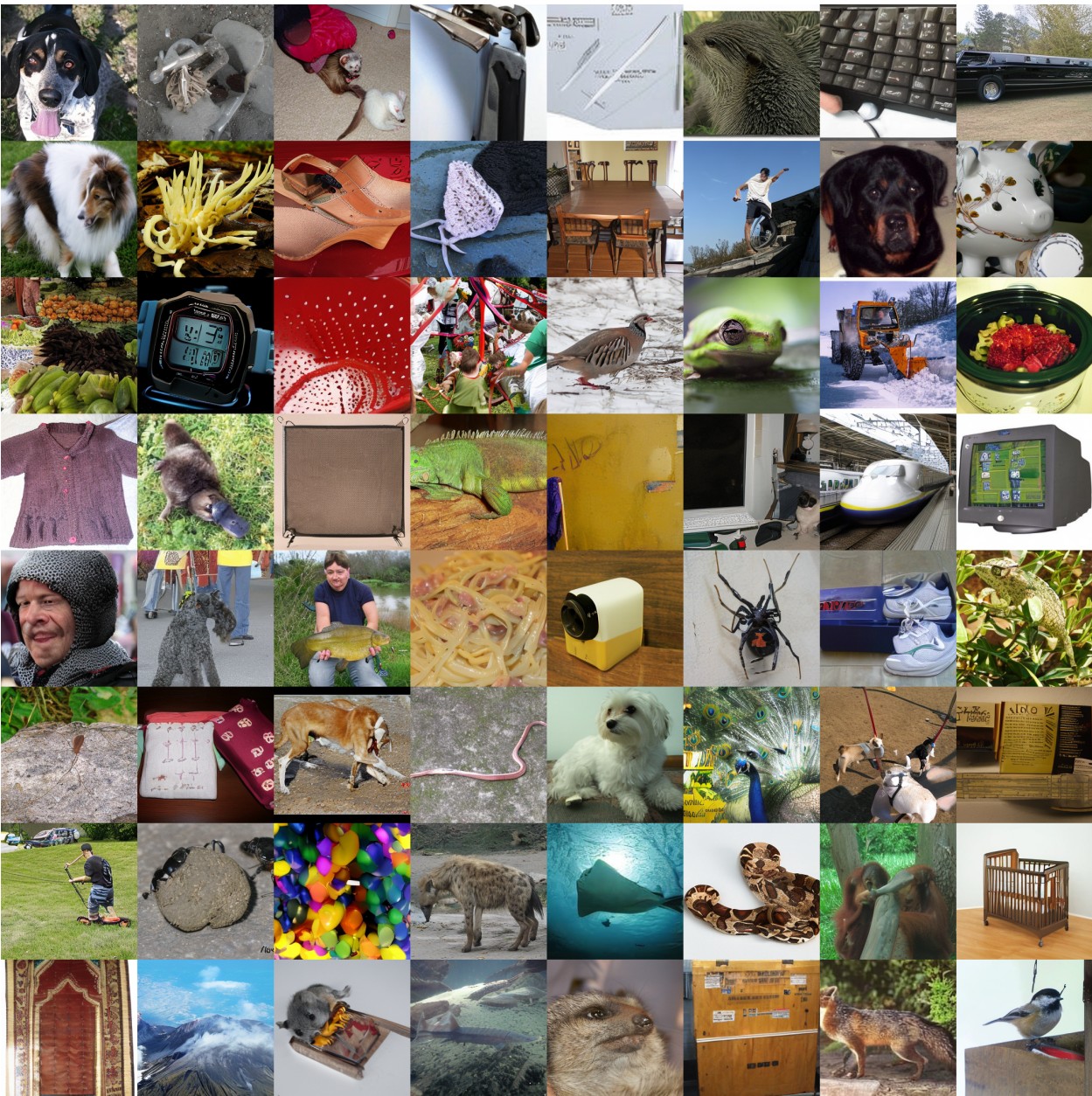

Figure 9: **Qualitative Comparison.** The generation results from REG (SiT-XL) with FPO on the ImageNet dataset with CFG.

