# OpenReview forum: "REPA-FPO: A Fisher Policy Optimization for Efficient Flow Matching Training"
_TMLR — Decision pending for TMLR_

### Review · Reviewer_14Pd · 2026-05-10

**Summary Of Contributions:**

This paper introduces a novel method to improve the training of flow matching models. The main idea is to move from uniform sample importance within a batch to weighting samples according to the empirical Fisher information. Since the empirical Fisher information is hard to compute directly, the authors instead use the loss magnitude as a proxy.
The authors demonstrate the applicability of their idea across different baseline methods by adding their sample weighting mechanism on top. They also provide an interesting theoretical motivation for their method.

I think the paper is in general "good" and I have only minor comments mentioned below.

**Audience:**

Yes

**Audience Explanation:**

The topic is of general relavence for anyone using flow matching and potentially beyond.

**Claims And Evidence:**

Yes

**Claims Explanation:**

The authors derive their results from a theoretical perspective + provide experiments that demonstrate that their wieghting mechanism is beneficial.

**Requested Changes:**

I am not sure if the comparisons are all fair. Looking at Figure 2, I see that training is faster when adding sample weighting. However, we are usually more interested in final performance after letting all methods converge, rather than stopping after x epochs.

I found the naming and framing of the method somewhat misleading. The connection to policy optimization in the RL sense seems loose, since the actual mechanism is a deterministic softmax-based reweighting of training samples by their loss magnitude.

I wonder: can the method also be applied to training settings that are not flow matching models?

It would be interesting to see standard errors over multiple runs, as the overall improvement from adding the method is small.

---

> ### Author Response · Authors · 2026-05-22
>
> We sincerely thank the reviewer for the encouraging evaluation and the thoughtful comments. Below, we carefully address each comment.
>
> Q1. Fairness of comparison — final performance after convergence
>
> Thank you for raising this important point. We completely agree that final converged performance is more meaningful than early-stage speed alone. We note that **400K iterations is a standard benchmark checkpoint** widely adopted by existing methods (e.g., REPA, REG, etc) for fair comparison on ImageNet-256, which is why we report main results at this setting. Beyond this, we also verify the effectiveness of FPO under extended training. In **Table 3** of the manuscript, we train the state-of-the-art REG framework with FPO for more iterations: FPO achieves an FID of **1.46 at only 800K** iterations, outperforming the vanilla REG baseline trained for **1.5M** iterations (FID 1.48). Notably, an FID of 1.46 on ImageNet-256 is already near the performance ceiling, indicating that FPO has almost converged at this point.
>
> Q2. Naming — connection to policy optimization in the RL sense
>
> We appreciate this observation and acknowledge that the term policy optimization may indeed cause confusion with RL methods. We want to clarify that the FPO policy refers to an adaptive sample-level weight allocation strategy — it dynamically determines how gradient budgets are distributed across samples within each training batch. This is closer to a resource allocation policy rather than an RL policy over sequential actions. We thank the reviewer for pointing out this potential issue and add a clarifying remark in the revised paper to make this distinction explicit. (Please see **Section 3.3**, highlighted in red.)
>
> Q3. Applicability beyond flow matching
>
> Thank you for this insightful question. We believe FPO can be applied beyond flow matching. The theoretical analysis of FPO (Proposition 3.1) relies on an MSE-type loss under a Gaussian observation model, which is not specific to flow matching. We provide empirical evidence supporting this:
>
> + **Table 9** demonstrates that FPO is effective under **noise prediction** with DiT on FFHQ, which is a different prediction target from the velocity prediction used in flow matching.
> + **Table 10** shows that FPO improves **vanilla flow matching** (trained without REPA) on CelebA-HQ.
>
> These results suggest that FPO generalizes across different prediction parameterizations and training frameworks.
>
> These results are presented in **Appendix C** of the manuscript.
>
> Q4. Standard errors and magnitude of improvement
>
> Thank you for this suggestion. We provide multi-seed evaluation in **Table 12** on REPA-XL with CFG, which shows that FPO achieves lower mean FID with smaller variance across different seeds, confirming the consistency of the improvement. Regarding the magnitude, we would like to offer two considerations: (i) At the current SOTA level (FID < 2), the performance gap between methods is naturally small, and we believe the consistent improvement across multiple settings and seeds is still meaningful. (ii) From a training efficiency perspective, as shown in **Table 3**, FPO achieves an FID of 1.46 at 800K iterations, surpassing the baseline trained for 1.5M iterations (FID 1.48), which corresponds to a $1.88\times$ reduction in training cost. We believe this efficiency gain, combined with the negligible overhead of FPO, makes it a practical and beneficial plug-in for existing frameworks.
>
> We hope the above responses address the reviewer's concerns. We are grateful for the constructive feedback and incorporate the suggested clarifications in the revised manuscript. We remain happy to address any further questions.

---

> > ### Comment · Reviewer_14Pd · 2026-05-22
> >
> > Thank you for the clarifications. I think the paper improved and is valuable for the community

---

> > > ### Author Response · Authors · 2026-05-22
> > >
> > > Thank you very much for the positive feedback and for the valuable suggestions that helped improve our paper. We are glad that the revisions address your concerns, and we appreciate your recognition of the contribution.

---

### Review · Reviewer_VaYK · 2026-05-14

**Summary Of Contributions:**

This paper proposes Fisher Policy Optimization (FPO), a dynamic sample reweighting strategy for training Flow Matching models. The core idea is to use the per-sample loss magnitude as an efficient proxy for the trace of the empirical Fisher Information Matrix (FIM), which then guides a gradient redistribution policy that up-weights high-information samples and down-weights low-information ones. The authors provide theoretical justification (Propositions 3.1 and 3.2) linking loss magnitude to FIM trace, and derive gradient dynamics (Theorem 3.3) showing that FPO introduces an additional weight-gradient term that vanishes near convergence. Extensive experiments on ImageNet with REPA and REG architectures demonstrate consistent improvements in both training efficiency (e.g., 1.25×–1.88× speedup) and generation quality (e.g., FID improvement from 24.40 to 22.50 on REPA-B). The method is also shown to be orthogonal to existing techniques and robust across different samplers and model scales.

**Audience:**

Yes

**Audience Explanation:**

The topic is important

**Claims And Evidence:**

Yes

**Claims Explanation:**

The connection between loss magnitude and FIM trace is well-derived and empirically validated, making the approach computationally feasible for large-scale training.

**Requested Changes:**

The authors may need to mention the work of Zhang et al. (2026) on effective policy learning for online coordination beyond submodular objectives. While their setting (multi-agent coordination) differs from Flow Matching training, their method for handling non-submodular sample/agent contributions and dynamically learning coordination policies offers a complementary perspective on sample-wise importance. Citing this work would enrich the positioning of FPO by situating it within a broader context of adaptive optimization and policy learning for sample efficiency.

While the ablation studies show robustness, the authors should provide more guidance on selecting the retention ratio and augmentation factor for new datasets or model scales. Currently, the optimal values are tuned on ImageNet; it would be helpful to comment on whether these transfer to other domains (e.g., text-to-image or video).

Zhang, Qixin, et al. "Effective Policy Learning for Multi-Agent Online Coordination Beyond Submodular Objectives." Advances in Neural Information Processing Systems 38 (2026): 164278-164339.

---

> ### Author Response · Authors · 2026-05-22
>
> We sincerely thank the reviewer for the positive evaluation and the constructive suggestions. We are glad that the reviewer recognizes the theoretical motivation and the practical effectiveness of FPO. Below, we carefully address each comment.
>
> Q1. Missing related work on adaptive policy learning.
>
> Thank you for pointing out this relevant work. We agree that adaptive policy learning for handling heterogeneous contributions is a broader theme that extends beyond generative modeling. We add a discussion of Zhang et al. [1] in the revised Related Work section. (Please see Section 2, highlighted in red.)
>
> ​	[1] Effective Policy Learning for Multi-Agent Online Coordination Beyond Submodular Objectives. NeurIPS-2025
>
> Q2. More analysis and discussion on hyperparameter selection and generalizability.
>
> Thank you for this constructive suggestion. We acknowledge that the optimal hyperparameters ($r$ and $\beta$) may vary across different settings. However, FPO introduces only two key hyperparameters, and as shown in **Table 6**, the method is robust across a wide range of values, consistently outperforming the baseline.
>
> Regarding generalizability, we validate FPO across diverse hyperparameter ranges, model architectures, datasets, diffusion spaces, and prediction targets. To further address the reviewer's concern, we conduct additional t2i experiments on **COCO** with **U-ViT**. The results are as follows:
>
> | Method | Iterations | FID↓ |
> |--------|-----------|------|
> | U-ViT (Baseline) | 1000K | 5.95 |
> | U-ViT + FPO | 800K | 5.82 |
>
> FPO achieves a lower FID of **5.82 at 800K** iterations, outperforming the baseline trained for **1000K** iterations (FID 5.95), further demonstrating its generalizability. We add these results to the revised paper. (Please see **Appendix C**, highlighted in red.) We thank the reviewer for highlighting video generation as an important direction. However, training video generation models from scratch requires substantial computational resources, and we leave this exploration as future work.

---

### Review · Reviewer_Vd56 · 2026-05-21

**Summary Of Contributions:**

This paper proposes a dynamic sample reweighting strategy for training REPA models. It uses the per-sample loss magnitude as an efficient proxy for the trace of the empirical Fisher Information Matrix (FIM) and leverages it to redistribute gradients in each training iteration. The paper presents extensive quantitative and qualitative experiments across various diffusion frameworks, all demonstrating improved training efficiency and generation quality without introducing additional bias. These results indicate that the proposed method is highly generalizable and can effectively enhance the training of diffusion models.

**Audience:**

Yes

**Audience Explanation:**

Training diffusion models is a key topic in generative modeling research, and finding efficient ways to train them is of great importance to the community.

**Claims And Evidence:**

Yes

**Claims Explanation:**

In the paper, the authors present the motivation behind the method’s design and provide empirical validation in Figures 3, 4 and Table 7, which effectively support this design. Furthermore, evaluations across various performance metrics indicate that the method does not introduce additional bias.

**Requested Changes:**

- 1. The authors should discuss or clarify the differences between the FPO strategy proposed in this paper and policy optimization in reinforcement learning. As currently understood, the FPO strategy appears to be inspired by reinforcement learning; therefore, it is recommended that the authors explicitly clarify its distinction from reinforcement learning within the paper.

- 2. The experiments in this paper primarily focus on the label-to-image generation task; the authors may consider extending them to text-to-image generation tasks. Such an extension would not only further substantiate the effectiveness of the method but also help validate the generalizability of FPO across different modalities.

---

> ### Author Response · Authors · 2026-05-22
>
> We sincerely thank the reviewer for the positive evaluation and the valuable suggestions. Below, we address each comment.
>
> Q1. Clarify the distinction between FPO and policy optimization in reinforcement learning
>
> Thank you for this suggestion. We agree that the term policy optimization may cause confusion with RL methods. We want to clarify that the FPO policy refers to an adaptive sample-level weight allocation strategy that dynamically distributes gradient budgets within each training batch, which is closer to a resource allocation policy rather than an RL policy over sequential actions. We add a clarifying remark in the revised paper. (Please see Section 3.3, highlighted in red.)
>
> Q2. Extending experiments to text-to-image generation.
>
> Thank you for this constructive suggestion. We conduct additional text-to-image experiments on COCO with the U-ViT architecture. FPO achieves an FID of 5.82 at 800K iterations, outperforming the baseline trained for 1000K iterations (FID 5.95), demonstrating the generalizability of FPO across different modalities. We add these results in the revised paper. (Please see Appendix C, highlighted in red.)
> We hope the above responses address the reviewer's concerns. We are grateful for the constructive feedback and incorporate the suggested changes in the revised manuscript. We remain happy to address any further questions.

---

> > ### Comment · Reviewer_Vd56 · 2026-05-26
> >
> > I thank the authors for their detailed response, which has effectively addressed my concerns.

---

> > > ### Author Response · Authors · 2026-05-26
> > >
> > > We thank the reviewer for the constructive comments, which have helped us significantly improve our manuscript. We truly appreciate your time and effort.

---

### Decision · Action_Editor_ESw1 · 2026-06-28

**Recommendation:** Accept as is

**Audience:**

Yes

**Audience Explanation:**

Flow matching is currently one of the leading paradigms in generative modeling, with wide-ranging applications across multiple domains. Because training flow matching models is computationally expensive and typically requires large-scale datasets, any theoretically sound method that meaningfully improves training efficiency, such as FPO (this submission), will be of significant interest to the generative AI and broader machine learning communities within TMLR's audience.

**Claims And Evidence:**

Yes

**Claims Explanation:**

To improve the training of flow matching models, this work introduces Fisher Policy Optimization (FPO), a strategy that dynamically reweights training samples based on their Fisher Information, which is efficiently estimated via their loss magnitudes. The claims are well-supported by both theoretical derivations and empirical results:

* The authors provide rigorous mathematical analysis proving that loss magnitude serves as an effective proxy for the trace of the Fisher Information Matrix (FIM).

* Extensive experiments on ImageNet using various state-of-the-art architectures (e.g., REPA, REG) demonstrate consistent improvements in both training efficiency and generation quality.

During the rebuttal phase, reviewers raised valid concerns regarding: (a) missing discussions on related work in adaptive policy learning, (b) guidance on hyperparameter selection, (c) potential terminological confusion between FPO and reinforcement learning, and (d) the method's applicability beyond class-conditional ImageNet generation.

The authors provided a comprehensive rebuttal that effectively assuaged these concerns. Notably, they clarified the naming convention, added the missing literature, and provided additional text-to-image generation experiments on the COCO benchmark using the U-ViT architecture, which further reinforced the generalizability of the proposed method. Ultimately, all three reviewers agreed that the submission's claims are accurate and well-supported, recommending acceptance. The Action Editor has reviewed the submission, the reviews, and the author rebuttal, and concurs with this recommendation.